# Sentinel-1 time series for mapping snow cover depletion and timing of snowmelt in Arctic periglacial environments: Case study from Zackenberg and Kobbefjord, Greenland

Sebastian Buchelt[1], Kirstine Skov[2], Kerstin Rasmussen[3], and Tobias Ullmann[1]

[1]Department Physical Geography, Institute of Geography and Geology, University of Wuerzburg, 97074 Wuerzburg, Germany
[2]Department of Bioscience, Arctic Research Center, Aarhus University, 4000 Roskilde, Denmark
[3]Asiaq, Nuuk, Greenland

**Correspondence:** Sebastian Buchelt (sebastian.buchelt@uni-wuerzburg.de)

**Abstract.** Snow cover (SC) and timing of snowmelt are key regulators of a wide range of Arctic ecosystem functions. Both are strongly influenced by the amplified arctic warming and essential variables to understand environmental changes and their dynamics. This study evaluates the potential of Sentinel-1 (S-1) synthetic aperture radar (SAR) time series for monitoring SC depletion and snowmelt with high spatiotemporal resolution to capture their understudied small-scale heterogeneity. We use 97 dual-polarized S-1 SAR images acquired over north-eastern and 94 over south-western Greenland in the interferometric wide swath mode from the years 2017 and 2018. Comparison of S-1 intensity against SC fraction maps derived from orthorectified terrestrial time lapse imagery indicates that SAR backscatter can increase before a decrease of SC fraction is observed. Hence, increase of backscatter is related to changing snowpack properties during the runoff phase as well as decreasing SC fraction. We here present a novel empirical approach based on the temporal evolution of the SAR signal to identify start of runoff (SOR), end of snow cover (EOS) and SC extent for each S-1 observation date during melt using backscatter thresholds as well as the derivative. Comparison of SC with orthorectified time lapse imagery indicate that HV polarization outperforms HH when using a global threshold. The derivative avoids manual selection of thresholds and adapts to different environmental settings and seasonal conditions. With a global configuration (Threshold: 4 dB; polarization: HV) as well as with the derivative, the overall accuracy of SC maps was in all cases above 75 % and in more than half cases above 90 %. Based on the physical principle of SAR backscatter during snowmelt, our approach is expected to work well in other low vegetated areas and, hence, could support large-scale SC monitoring at high spatiotemporal resolution (20 m, 6 days) with high accuracy.

## 1 Introduction

### 1.1 Snow in arctic environments

Snow cover (SC) has been identified as an essential climate variable (GCOS-WMO, 2020) covering about 40 to 50 % of the northern hemisphere during winter (Dietz et al., 2012; Rees, 2006; Tsai et al., 2019b). It plays an important role for various components of the Earth system like hydrology, ecology and climatology as well as global energy, water and carbon cycles due to the large seasonal variability in snow extent on the northern hemisphere (between 4 to 46 million km$^2$ in summer and

winter, respectively (Rees, 2006)) and the specific physical properties (Arslan et al., 2017; Box et al., 2019; Dietz et al., 2012; Pedersen et al., 2018). Its high albedo (0.8 to 0.9) compared to snow-free coverages strongly influences the energy balance. Moreover, its insulating properties limit heat exchange between soil and atmosphere and, thereby, regulate the seasonal active layer thickness (Rees, 2006; Tsai et al., 2019b). The snowpack plays an important role for water storage and supply (Dietz et al., 2012; Marin et al., 2020) and it is a key factor for arctic phenology, ecology and the distribution of flora (Assmann et al., 2019; Ide and Oguma, 2013; Kepski et al., 2017). On the one hand, the thermal insulation protects plants from frost damages and the snowpack provides water and, with it, nutrients for the plants. On the other hand, SC blocks the sunlight needed for photosynthetic activity. Further, the metabolic activity of plants is directly linked to the timing of snowmelt (Assmann et al., 2019; Ide and Oguma, 2013; Kankaanpää et al., 2018; Kepski et al., 2017; Pedersen et al., 2018). However, the timing of snowmelt is highly variable in space and time and influenced by snow accumulation, redistribution, and ablation. The former two depend on the climatic conditions, e.g. latitudinal and altitudinal position and continentality, as well as on the local topography that affects transport of snow due to wind and gravitational redistribution. Thereby, snow is shifted from windward slopes, ridges, steep and high terrain to wind sheltered leeward slopes, sinks and low-lying terrain (Elberling et al., 2008; Farinotti et al., 2010; Lehning et al., 2008; Mott et al., 2018; Pedersen et al., 2016). Therefore, the redistribution effects generate a similar pattern of snow accumulation at the end of the winter primarily driven by the local topography, while the overall amount of snow depends on the amount of solid winter precipitation (Buus-Hinkler et al., 2006; Farinotti et al., 2010; Ide and Oguma, 2013; Kepski et al., 2017; Pedersen et al., 2018). Ablation is driven by temperature, turbulent fluxes and solar radiation (Mott et al., 2011, 2013), which, in combination with redistribution, leads to a high small-scale heterogeneity of SC during melt season (Mott et al., 2018). Knowledge about SC is important because it is decreasing with rising temperatures, resulting in a negative self-strengthening feedback between temperature and SC, which might partially drive the arctic amplification (Hock et al., 2019; Meredith et al., 2019; Rees, 2006). The decrease of SC duration and extent has been documented for the northern hemisphere over the last 40 years (Box et al., 2019; Brown and Robinson, 2011; Meredith et al., 2019) and could have a negative impact on species richness (Niittynen et al., 2018), but changes on local scale do not indicate clear trends (e.g. Pedersen et al., 2016; Young et al., 2018). Moreover, Hock et al. (2019) state that knowledge about SC distribution is still limited, especially at small spatiotemporal scales.

## 1.2 SAR remote sensing of snow

As such, SC monitoring requires remote sensing products at very high spatial and temporal resolution. Until recently, freely available synthetic aperture radar (SAR) data could not fulfil these requirements; however, as the Sentinel-1 (S-1) mission with its twin satellites provides freely available C-Band SAR imagery at a spatial resolution of 10–20 m and with a six-day repeat cycle, SAR data has become an attractive alternative (ESA, 2012; Marin et al., 2020). While ground-based methods lack in spatial coverage, established SC mapping methods based on optical spaceborne earth observation suffer from reduced temporal resolution due to cloud coverage and sun illumination, e.g. during polar night (Dong, 2018; Portenier et al., 2020; Tsai et al., 2019b). SAR remote sensing can overcome these limitations, as it operates independent of sun illumination and atmospheric conditions (Marin et al., 2020; Tsai et al., 2019b; Ulaby et al., 2014). Previous studies applying SAR data have indicated its

ability to detect wet snow, as liquid water in the snowpack decreases the dielectric constant leading to a higher absorption coefficient (Marin et al., 2020; Nagler and Rott, 2000; Ulaby et al., 2014). Hereby, the penetration depth decreases to a few centimetres and most of the signal is absorbed or reflected by the uppermost parts of the snowpack resulting in a strong decrease

of the backscatter (Marin et al., 2020; Ulaby et al., 2014). This effect is the basis for the bi-temporal approach *Nagler's method* (Nagler and Rott, 2000), which compares backscatter during snowmelt with reference satellite imagery acquired during snow-free or dry snow conditions (Nagler et al., 2016, 2018; Nagler and Rott, 2000; Snapir et al., 2019). However, this approach is only able to detect wet snow. More advanced methods additionally use digital elevation models (DEMs) (Nagler and Rott, 2000; Storvold and Malnes, 2004; Thakur et al., 2017) or optical remote sensing data (Nagler et al., 2018; Snapir et al., 2019;

Thakur et al., 2018) to monitor dry snow. Nevertheless, these approaches cannot cover the small-scale heterogeneity of SC, because DEM-assisted approaches overestimate dry snow in steep terrain, while the inclusion of optical remote sensing data decreases the spatiotemporal resolution of the product (Solberg et al., 2010; Storvold and Malnes, 2004; Tsai et al., 2019b).

According to Tsai et al. (2019b), SAR time series have rarely been investigated for SC mapping as most studies apply their method only on a few scenes and mostly for one single year, hence not taking advantage of the abundant data available. Only

70 Tsai et al. (2019a, c) developed a SC monitoring approach using the entire temporal information of a S-1 SAR time series by incorporating interferometric, polarimetric and backscatter features as well as elevation and land cover information into a supervised classification approach. On the contrary, recent studies developed methods for snow depth estimation (Lievens et al., 2019) and snowmelt phase detection (Marin et al., 2020) based only on the distinct seasonal SAR backscatter signal of snow

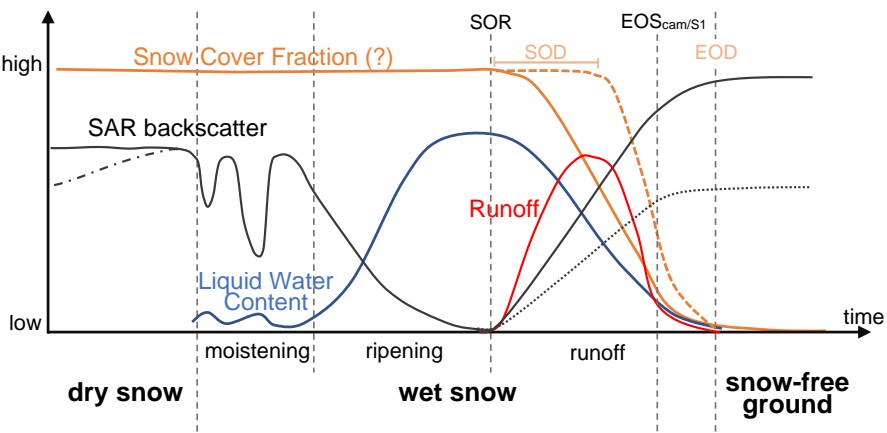

**Figure 1.** Schematic illustration of the pixel-wise seasonal evolution of synthetic aperture radar (SAR) backscatter intensity above snow with its phases of melt and relevant snow properties, i.e. liquid water content, runoff and snow cover fraction. This illustration (modified from Tsai et al., 2020) is based on the findings of Marin et al. (2020) and Tsai et al. (2020) and shows the parameters used in this study: SOR: start of runoff; SOD: start of snow cover fraction decrease which varies in time (indicated by the range of SOD and the different snow cover fraction curves) depending on snow pack properties and melt rates; $EOS_{cam/S1}$: end of snow cover detected by camera/Sentinel-1; EOD: end of snow cover fraction decrease. The understudied interaction between Sentinel-1 SAR backscatter and snow cover fraction during the runoff phase and the potential of SAR time series to detect snow cover depletion during melt are investigated by this study.

described in the following (Fig. 1): During winter, the liquid water content is close to 0 % and the snowpack is nearly transparent to C-Band SAR, unless for larger snowpacks (>1 m), where a sensitivity of C-band cross-polarization is indicated by Lievens et al. (2019, 2021). The returned signal is, therefore, predominantly scattered from ground surface underneath the snowpack with backscatter similar to snow-free conditions (Marin et al., 2020; Nagler et al., 2016). With increasing temperatures snow is wetting leading to a strong decrease in backscatter during the moistening and ripening phase (Marin et al., 2020; Nagler et al., 2016; Rees, 2006). In the last phase of snowmelt, which is called runoff, backscattering increases again until an intensity level comparable to snow-free ground conditions is reached (Marin et al., 2020). This steady increase is not fully understood yet but three possible explanation are described by Marin et al. (2020): (i) the increasing surface roughness of the snowpack; (ii) an increased number and size of intrusions like ice lenses or snow grains; (iii) subpixel parts turning into snow-free state leading to patchy SC towards the end of the melting period causing mixed pixel signal response (Marin et al., 2020).

We here propose a novel approach, which adapts *Nagler's method* (Nagler and Rott, 2000) to the seasonal evolution of the SAR signal using thresholds based on the seasonal minima of the SAR time series as well as the backscatter derivative for fast, simple, but effective SC mapping during snowmelt. With this new methodology, we can derive start of runoff (SOR) (based on the method from Marin et al. (2020)), end of snow cover (EOS), the extent of end-of-season SC and SC extent maps for each S-1 observation date during melt. We validate EOS and SC derived from S-1 time series (2017 & 2018) with a reference dataset, generated from time lapse photography available for Zackenberg Valley (Northeast Greenland) as well as Kobbefjord (Southwest Greenland) and assess the interaction between SC fraction and backscatter intensity in detail.

## 2 Study site and datasets

### 2.1 Study area

The Zackenberg (ZRA) and Kobbefjord (KRA) research areas are part of the Greenland Ecosystem Monitoring programme (www.g-e-m.dk), which has performed long-term monitoring of ecosystem components since 1995 (ZRA) and 2007 (KRA). The ZRA is located in high arctic Northeast Greenland (Fig. 2a) approx. 40 km west of the outer coast and 70 km east of the inland ice sheet (Meltofte and Rasch, 2008). ZRA covers most parts of the Zackenberg valley floor and the surrounding slopes of Zackenberg Mountain (west) and Aucellabjerg (north and east) (Fig. 2c). The mean annual temperature in the valley is about -9 °C and mean daily temperatures are usually above 0 °C from early June until mid-September (Hansen et al., 2008; Pedersen et al., 2018). The mean annual precipitation, with 80 to 85 % falling as snow, is about 260 mm but varies largely from year to year (150 – 400 mm) (Hansen et al., 2008). Maximum snow depths vary also considerably from year to year (0.4-1.4 m) (López-Blanco et al., 2020). Glacial ice occurs mostly at higher altitudes (>1000m) due to low precipitation (Mernild et al., 2007). The radiation budget is dominated by polar night and day, which have a length of 89 days and 106 days, respectively (Pedersen et al., 2016). The climate is rather continental with high temperature fluctuations and low humidity due to the build-up of sea ice during winter in the Young Sound (Westergaard-Nielsen et al., 2017). The topographic setting and predominant northern winds during winter cause similar patterns of snow accumulation every year (Elberling et al., 2008; Hinkler et al., 2008; Kankaanpää et al., 2018; Pedersen et al., 2016). Vegetation formations below 200 m a.s.l. are dominated by small shrubs

and grasses (Fig. 2e) (Elberling et al., 2008; Westergaard-Nielsen et al., 2017). With increasing altitude, the percentage of bare ground and rock increases, while above 600 a.s.l. only scarce vegetation is found (Buus-Hinkler et al., 2006; Elberling et al., 2008). The transition from SC to snow-free ground in ZRA begins in late May in years with low snow accumulation, but can be prolonged until early July in years with high snow accumulation and even until late July in the very extreme year of 2018 (López-Blanco et al., 2020). .

The KRA is located in low arctic Southwest Greenland at the bottom of the 16 km long fjord Kangerluarsunnguaq/Kobbefjord, approx. 20 km southeast of Greenland's capital, Nuuk (Fig. 2b). KRA drainage area, which covers 32 km$^2$, is characterized by three valleys surrounded by steep mountains up to 1375 m (Fig. 2d). Three major lakes (5 % of total area) dominate the valley system (Abermann et al., 2021). The GEM ClimateBasis main weather station (KOB) is located at the eastern end of the largest lake and provides high quality meteorological data. The mean annual air temperature in KRA at KOB is -0.1 °C (2008-2020), with highest monthly temperatures in July (10.6 °C) and lowest in February (-8.6°C). The mean annual precipitation is around 830 mm and about one third of it falls as snow (Abermann et al., 2021). Both total annual precipitation and maximum snow depths vary considerably from year to year (470-1170 mm and 0.3-1.3 m, respectively), and the winters are often characterized by shorter warmer periods causing episodic snow melt (Pedersen et al., 2015). The predominant winter wind directions (eastern sector) and the topography cause similar, yet highly heterogeneous patterns of snow accumulation every year (Myreng et al., 2020). The transition from snow covered to snow-free ground in KRA normally begins in late April or early May and the lower elevated parts of the valley system will usually be snow-free around the beginning of June. The rapid snowmelt often concurs with strong discharge peaks in the monitored rivers (Abermann et al., 2019). The KRA valleys are dominated by dwarf shrub heath, dry south-facing slopes and smaller fen areas (Bay et al., 2008), whereas the steep mountain slopes and higher elevated areas, are dominated by bare ground, snow patches, rockslides and small hanging glaciers (Pedersen et al., 2015).

Physical snow properties (e.g. SC fraction using time-lapse cameras) are measured regularly within the GeoBasis program at both ZRA and KRA (Skov et al., 2019; Westergaard-Nielsen et al., 2017; Rasmussen et al., 2020) following generally the same protocols, however being adapted to local conditions and logistics.

## 2.2  Datasets

For this study, we investigated all S-1 single look complex (SLC) data Interferometric Wide swath mode (IW) acquisitions from the ascending orbit (relative orbit 74) over ZRA and one descending orbit (relative orbit 54) over KRA between the 1 January 2017 and the 31 December 2018 from the European Space Agency (ESA) Copernicus Open Access Hub (ESA, 2021) and the Alaska Satellite Facility (Alaska Satellite Facility, 2021) at a repeat cycle of six days. As dual-polarimetric mode (HH + HV) started being operational for Zackenberg (orbit 74) only from 21 May 2017 onwards, all previous acquisitions were neglected leading to a limited time series in 2017 (ESA, 2020). In total, 38 scenes in the year 2017 and 59 scenes in the year 2018 were downloaded for ZRA as well as 41 (2017) and 53 (2018) scenes for KRA. The local acquisition time is 18:30 UTC in the afternoon in ZRA and 6:45 in the morning (9:45 UTC) in KRA.

In ZRA, daily time lapse imagery of the central part of the Zackenberg Valley has been taken from the east facing slope of the Zackenberg mountain at solar noon since 1997 (Skov et al., 2019; Buus-Hinkler et al., 2006). The selected camera field of

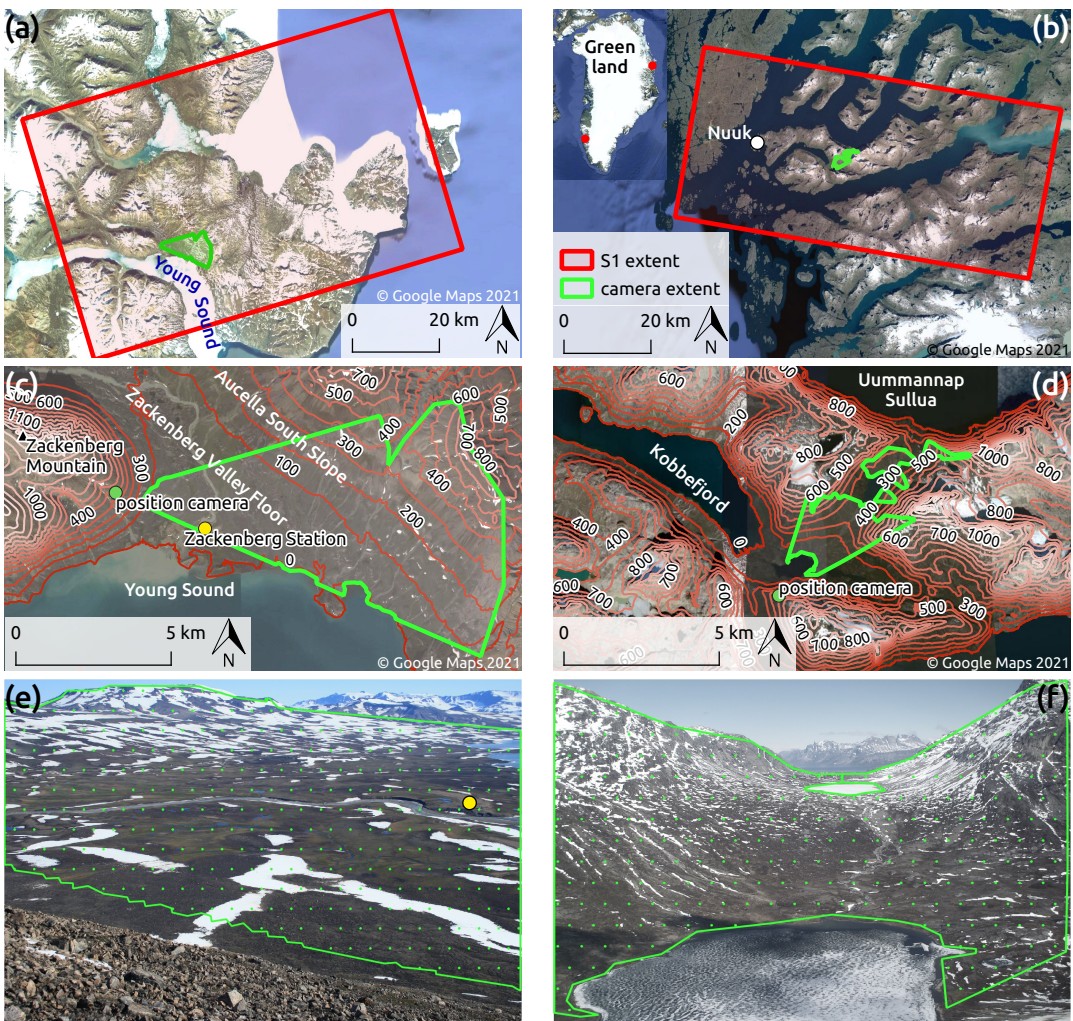

**Figure 2.** (a/b) Overview of Zackenberg/Kobbefjord study area and Sentinel-1 scene located in northeastern/southwestern Greenland (see red dots in inlet of (b)). (c/d) Study sites of Zackenberg (c) and Kobbefjord (d) with camera field of view and location of time lapse camera (green dot) and Zackenberg Research Station (yellow dot). (e) Master image of time lapse camera with used field of view and location of Zackenberg station (yellow dot). Background image from GeoBasis Zackenberg (Greenland Ecosystem Monitoring Secretariat, 2021). (f) Master image of Kobbefjord study area. Background image from GeoBasis Kobbefjord (Greenland Ecosystem Monitoring Secretariat, 2021).

view excludes far distant ranges and close foreslopes. It covers about 45 km$^2$ of the valley floor and the west-facing slopes of Aucellabjerg (see Fig. 2c,e). The time lapse imagery from 2017 and 2018 was taken with a 10 Megapixel Canon EOS 1000D camera system and stored in a 24-bit JPG format (Skov et al., 2019; Westergaard-Nielsen et al., 2017). In KRA, daily images cover two lakes, the valley floor and the surrounding slopes. The selected field of view excludes hidden slopes in far range as well as the lake surface and covers about 7 km$^2$. Images are acquired by an 6 Megapixel HP Photosmart E427 camera system

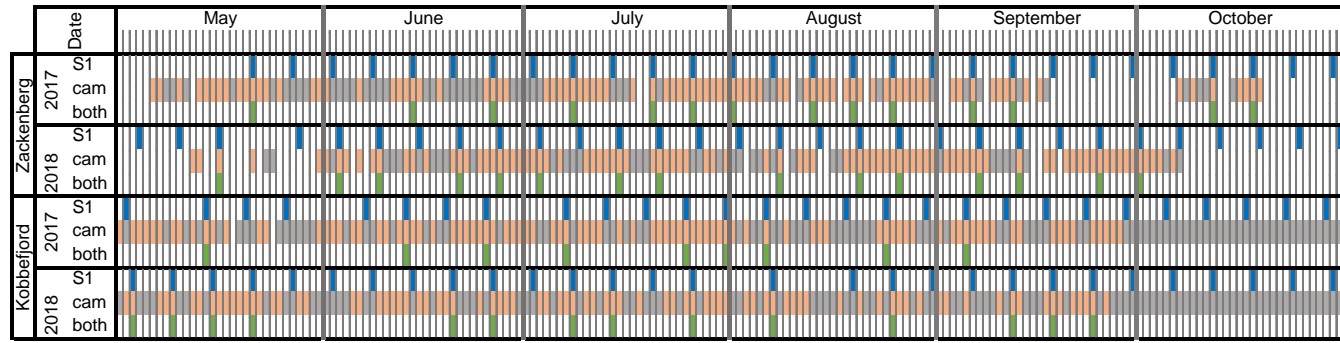

**Figure 3.** Acquisition dates of available data: blue: Sentinel-1 (S-1); grey: camera time lapse imagery excluded due to bad quality; orange: used camera images; green: coinciding S-1 and camera dates, which were used for comparison and assessment of the products. The developed approach also uses S-1 acquisitions before May and after October, but are not shown here for better visualization of the relevant observation period.

in JPG format (Rasmussen et al., 2020). We manually excluded images of bad quality due to cloud cover, fog, or rain prior to the processing. In ZRA, 89 out of 137 images in 2017 and 91 out of 127 in 2018 were found useful (Fig. 3). In KRA, images before 1 May and after 30 September were also excluded due to mountain shadow, but start-of-season SC extent was visually cross-checked to ensure the correct SC area is covered. 93 out of these 151 images in 2017 and 84 out of 153 images in 2018 were found useful (Fig. 3). The Greenland Ecosystem Monitoring program GeoBasis (Greenland Ecosystem Monitoring Secretariat, 2021) provide measures on SC fraction in ZRA derived from the above-mentioned camera (Skov et al., 2019; Westergaard-Nielsen et al., 2017), which is used as additional validation of the reference SC maps generated for this study.

Further, we used the ArcticDEM with a spatial resolution of 2 m (Porter et al., 2018; Noh and Howat, 2015; Polar Geospatial Center, 2021) for the processing of the S-1 scenes and the camera images, as it provides sufficiently high spatial resolution and accuracy needed for the calibration and orthorectification (Candela et al., 2017; Meddens et al., 2017). High resolution optical data from PlanetScope (Planet Team, 2021) satellites with a spatial resolution of 3 m are additionally used for the orthorectification process of the time lapse imagery. We selected for both sites a single cloud-free PlanetScope acquisition during the snowmelt season of 2018 with many distinguishable snow patches and patterns for which a corresponding time lapse camera image of good quality was available. The used scene in ZRA (Satellite IDs: 1008 & 1014) was acquired on the 12 August 2018 at about 11 AM local time. The scene for KRA (Satellite ID: 0f2d) was acquired on 15 June 2018 1 PM UTC (10 AM local time).

## 3 Methodology

### 3.1 Sentinel-1 snowmelt and snow cover products

The processing of S-1 products followed the workflow developed in Ullmann et al. (2019): The scenes were processed in IDL (version 8) and ENVI (version 5) using the functionalities of the ESA SNAP software (version 7). The processing included the application of the most recent orbit-file, split (ZRA: IW2, bursts 3 to 5; KRA: IW3, bursts 6 to 7), calibration to backscatter coefficient $\beta_0$ and deburst. Thereafter, multi-looking (3 looks in range, 1 look in azimuth), speckle filtering using a boxcar filter with a window size of 3 by 3 pixels and calibration to backscatter coefficient $\gamma_0$ using the terrain flattening approach similar to Small (2011) (Small et al., 2021) with the ArcticDEM (Porter et al., 2018) was applied. The data were terrain corrected using the Range-Doppler approach (Richards, 2009; Ulaby et al., 2014) and the ArcticDEM (Porter et al., 2018). Areas of shadow and layover were masked out. Datasets were processed to a spatial resolution of 20 by 20 m and all scenes were stacked and resampled to a common grid using bi-linear interpolation. The final product of S-1 preprocessing is the temporal stack of $\gamma_0$ backscatter intensities in decibel (dB) for both, HV and HH polarizations (Ullmann et al., 2019).

Our approach uses the $\gamma_0$ backscatter intensity time series and a simple set of thresholds as input to (i) identify day of year (DOY) of start of runoff (SOR) and end of snow cover (EOS), (ii) detect start-of-season snow-free areas and end-of-season snow-covered patches and (iii) derive a SC extent map for each S-1 observation date during melt (Fig. 4). The threshold setting is based on the characteristic seasonal backscatter behaviour above snow, which is described by Marin et al. (2020) and Lievens et al. (2019) as a strong decrease in intensity in early spring followed by an increase in intensity during melt in late spring and summer. To detect the seasonal development of intensities, we compute the seasonal pixel-wise backscatter minimum $min(\gamma_0)$ during the melt period (1 March – 31 August) as an adaptive reference. Thereafter, $SOR_{S\text{-}1}$ is determined (Eq. (1)) as the day of year (DOY), where the backscatter intensity ($\gamma_0$) reaches the seasonal minimum $min(\gamma_0)$ (i.e. in accordance with runoff phase detection of Marin et al. (2020)):

$$
SOR_{S\text{-}1}(x,y) = \begin{cases} DOY & \text{, if } \gamma_0(x,y,DOY) = min(\gamma_0) \\ -1 & \text{, otherwise} \end{cases} \tag{1}
$$

Following $SOR_{S\text{-}1}$, $EOS_{S\text{-}1}$ is identified using two different approaches: the threshold-based approach or the derivative-based approach. First, we describe the former: $EOS_{S\text{-}1}$ is determined as the DOY, where the backscatter exceeds the seasonal minimum $min(\gamma_0)$ by more than a user-defined threshold $t$ (Eq. (2)). Two further conditions are applied to filter events of melting followed by a freeze up of the snowpack, as these events cause a similar temporal evolution of the backscatter signal and, therefore, could lead to false detection: (i) In order to filter short-term snowpack refreeze events during the melt, the exceeding of $t$ must apply for three consecutive acquisitions (Eq. (2)). (ii) If backscatter values with less than 2 dB difference to the seasonal minimum are observed after a detected EOS and during early melt season (before 1 July), the pixel is considered as wet snow and a new EOS will be searched. Thereby, we filter early season short-term melt events followed by a refreeze of the snowpack. A pixel is classified as *start-of-season snow-free*, if no distinct temporal signal is found and the backscatter does

not exceed the threshold $t$ for three consecutive acquisitions (Eq. (2)).

$$EOS_{S\text{-}1}(x,y) = \begin{cases} \text{first } DOY & \text{, where } \gamma_0(x,y,DOY) > (min(\gamma_0)+t) \text{ (three consecutive times)} \\ \text{start-of-season snow-free} & \text{, otherwise} \end{cases} \tag{2}$$

Besides this threshold-based approach, we also implement an approach using the derivative of the backscatter time series to detect EOS. The derivative ($\Delta\gamma_0\_DOY(x,y)$) is defined as the difference of backscatter in dB between two acquisitions (Eq. (3)). We believe that the derivative will adapt automatically to different environmental and seasonal conditions and, hence, does not require an manual parametrisation of the threshold. As the derivative is more prone to speckle induced variations, we additionally filter the time series in the temporal domain with a window of four acquisitions (one before, two after). By

using an equal number of images, we reduce effects of systematic backscatter variations between S-1A & B. We do not apply a spatial filter to keep the same high spatial resolution as in the threshold approach.

$$\Delta\gamma_0(x,y,DOY) = \gamma_{0\_filtered}(x,y,DOY+\Delta t) - \gamma_{0\_filtered}(x,y,DOY) \tag{3}$$

Using the filtered time series, we first exclude areas that show less than 3dB variation during the melt period (March -August) and classify them as *start-of-season snow-free* as we assume that such areas do not show a distinct backscatter behaviour related

to snowmelt. Then, $EOS_{S\text{-}1}$ is determined as the DOY, where the derivative is below 1dB after the derivative has shown an increase larger than 1dB (Eq. (4)). If, thereafter, $\Delta\gamma_0\_DOY(x,y)$ exceeds 1dB at a higher backscatter level than the previously detected EOS, $EOS_{S\text{-}1}$ is reassigned as DOY, where the derivative falls again below 1dB.

$$EOS_{S\text{-}1}(x,y) = \begin{cases} \text{start-of-season snow-free} & \text{, variation below 3dB} \\ \text{first } DOY & \text{, where } \Delta\gamma_0(x,y,DOY) < 1dB \text{ (after } \Delta\gamma_0(x,y,DOY) > 1dB) \\ \text{start-of-season snow-free} & \text{, otherwise} \end{cases} \tag{4}$$

Due to the high arctic environment, end-of-season snow-covered patches can occur. However, the algorithms presented in

Eq. (2) and Eq. (4) to find EOS can only detect the end of the wet snow status, but cannot distinguish, whether the status changes to a snow-free or back to a dry-snow state. Therefore, we implement an additional condition to identify end-of-season snow-covered patches based on the following assumption: end-of-season snow-cover has undergone several melt- and refreeze-cycles and has strong structural similarity to firn which contains large snow grains and ice lenses that act as targets causing strong backscattering, especially volume scattering (Marin et al., 2020; Nagler and Rott, 2000). Therefore, the HV

backscatter in autumn and early winter, once the snowpack is completely refrozen, should be much higher than in areas with snow completely melted and now covered by new snow with small grains. Visual assessment of the HV backscatter confirms that this effect is occurring. Accordingly, pixels are classified as *end-of-season snow-covered*, if their maximum HV backscattering $max(\gamma_{0\_HV})$ in autumn (1 October – 31 December) exceeds $min(\gamma_0)$ by 9 dB and the detected EOS is after 15 August (i.e. DOY 227) (Eq. (5)). The threshold of 9 dB is set in accordance with our observations that HV snow-free summer

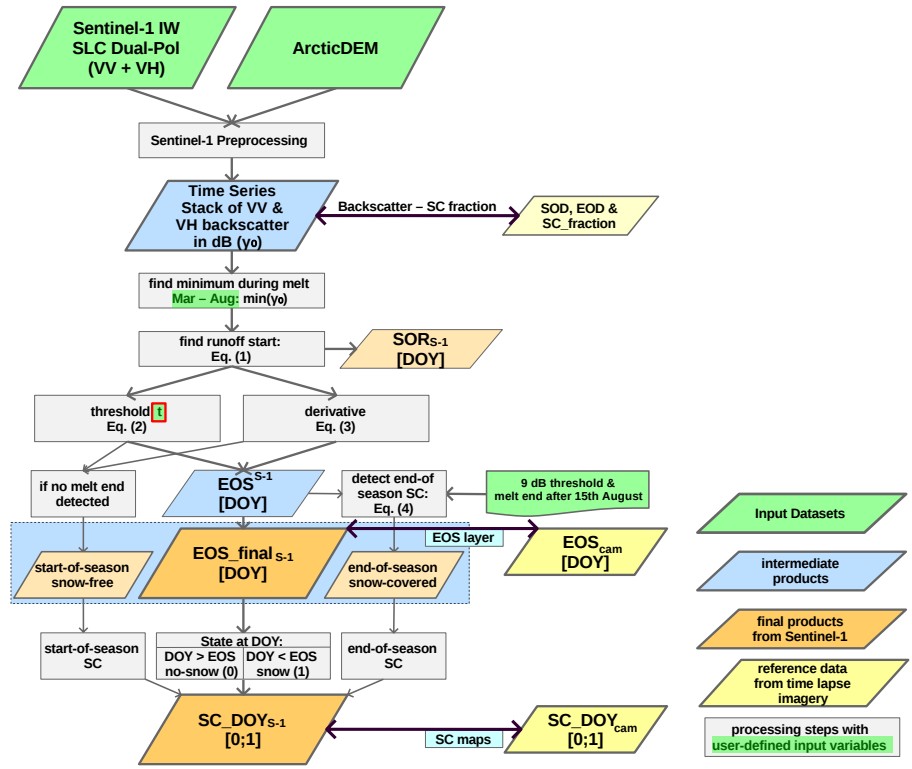

**Figure 4.** Workflow chart of proposed approach to detect (i) snow cover (SC) and timing of snowmelt as (ii) start of runoff (SOR) and (iii) end of snow cover (EOS) from Sentinel-1. EOS, (iv) start-of-season snow-free and (v) end-of-season snow-covered areas are visualized together as so-called melt layer (Fig. 11 & 12), whereas analysis is conducted separately.

intensity does not exceed the seasonal minimum by more than this value, but might not apply for other sites.

$$
EOS\_final_{S\text{-}1}(x,y) = \begin{cases} end\text{-}of\text{-}season\ snow\text{-}covered & \text{, if} \begin{pmatrix} max(\gamma_{0\_HV}) > (min(\gamma_0) + 9)\ \text{and} \\ EOS_{(S\text{-}1)}(x,y) > 227 \end{pmatrix} \\ EOS_{(S\text{-}1)}(x,y) & \text{, otherwise} \end{cases} \tag{5}
$$

As a final step, the following Eq. (6) is used to derive SC maps from the EOS, the *start-of-season snow-free* and the *end-of-season snow-covered* products:

$$
SC\_DOY_{S\text{-}1}(x,y) = \begin{cases} 0\ \text{(no-snow)} & \text{, if } DOY \geq EOS\_final_{S\text{-}1}(x,y)\ \text{or } start\text{-}of\text{-}season\ snow\text{-}free \\ 1\ \text{(snow-covered)} & \text{, if } DOY < EOS\_final_{S\text{-}1}(x,y)\ \text{or } end\text{-}of\text{-}season\ snow\text{-}covered \end{cases} \tag{6}
$$

The resulting binary SC map at a specific DOY is 1 (*snow-covered*) or 0 (*no-snow*), if the DOY is before or after EOS, respectively. Optionally, further information on the snow status could be derived from SOR/EOS and delineate, where SC is

contributing to runoff or not. For the further assessment of the S-1 SC products, this separation is neglected, as the terrestrial camera is not able to detect SOR.

## 3.2 Time lapse imagery snow cover products

Orthorectified SC maps are generated from time lapse camera images to validate the S-1 EOS and SC products (Fig. 5): (i) All time lapse camera images are aligned to the master image (12 August 2018 – ZRA; 15 June 2018 – KRA) using discrete fourier transformation (implemented in the python package *imreg_dft* (Týč and Gohlke, 2014)). (ii) The stacked images are classified using the histogram minimum thresholding approach of Salvatori et al. (2011) in the following way: All pixel above and below the minimum of a bimodal histogram distribution are assigned as *snow-covered* and *no-snow*, respectively. However,
we use the brightness histogram instead of the blue band to avoid misclassification of lakes and ponds reflecting the sky (like e.g. Westergaard-Nielsen et al., 2017). (iii) The master image is then orthorectified using the python package *georef_webcam* (Buchelt, 2020) based on the approaches of Corripio (2004), Härer et al. (2016) and Portenier et al. (2020). The ArcticDEM (Porter et al., 2018) is used for the projection and ground control points (GCPs) of remarkable landscape features are derived from same-day high-resolution PlanetScope imagery to optimize the orthorectification procedure resulting in a projection with
high geospatial accuracy. (iv) Thereafter, the projected master is coregistered to the PlanetScope image with additional GCPs. (v) Based on the georeferenced master, all aligned and classified images are projected to SC maps in the coordinate system of S-1 with a spatial resolution of 2.5 m and then aggregated to 20 m resolution by calculating the SC fraction ($SC\_Fraction_{DOY}$).

For comparison with the S-1 products, *start-of-season snow-free* areas, *end-of-season snow-covered* patches and EOS are derived from the SC fraction map using Eq. (7) (see also Fig. 5):

$$EOS_{camera}(x,y) = \begin{cases} start\text{-}of\text{-}season\ snow\text{-}free & \text{, if } SC\_Fraction_{DOY\_1} < 0.5 \\ DOY & \text{, where } SC\_Fraction_{DOY} < 0.5 \text{ (three consecutive times)} \\ end\text{-}of\text{-}season\ snow\text{-}covered & \text{, if } SC\_Fraction > 0.5 \text{ (for all acquisitions)} \end{cases} \quad (7)$$

(i) Pixels, which are less than 50 % snow covered in the first observation in spring ($SC\_Fraction_{DOY\_1}$), are classified as *start-of-season snow-free*, whereas (ii) *end-of-season snow-covered* are at least 50 % snow covered in all orthorectified SC fraction maps. (iii) EOS is determined for the remaining area as the first of three consecutive dates, where SC fraction falls below 50 %. Besides, (iv) Binary SC maps ($SC\_DOY_{camera}$) for validating S-1 SC maps are generated using the boundary
condition that pixels have to be more than 50 % snow covered in order to be classified as snow. (v) Further, DOY of start and end of SC decrease (SOD/EOD) are defined for each pixel as the last observation with 100 % SC fraction and the first observation with 0 % SC fraction, respectively. SOD and EOD are only used for the backscatter - SC fraction interaction analysis.

We tested the accuracy of the orthorectification procedure using 25 [26] GCPs to assure the correct location of the projected
maps (Fig. 6b). The root mean square error (RMSE) of the orthorectification was 9.4 m [5.3m] in Zackenberg [Kobbefjord], hence, less than one S-1 pixel (20 m). Besides, SC fraction maps from the time lapse imagery in Zackenberg are validated with not orthorectified in situ data which used the same time lapse images as source (Greenland Ecosystem Monitoring Secretariat,

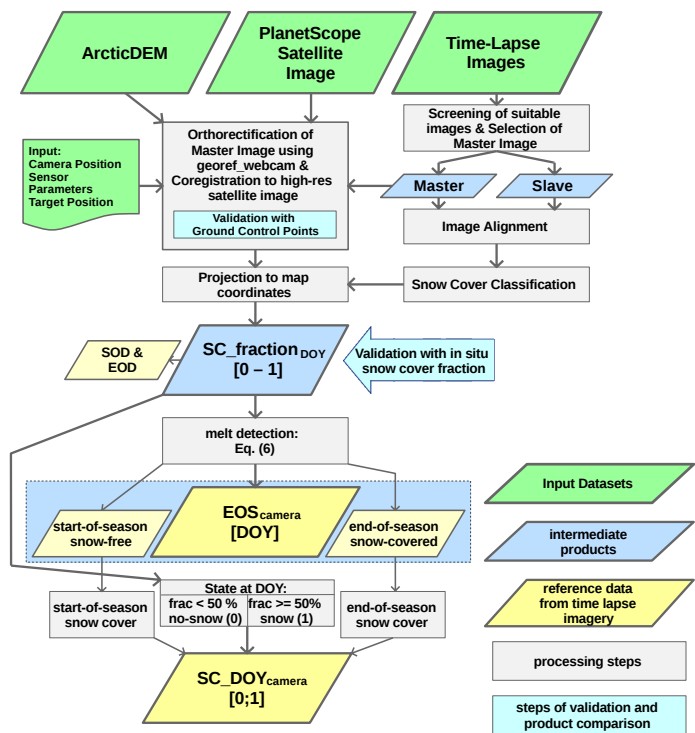

**Figure 5.** Workflow chart of proposed approach to detect (i) snow cover (SC) and timing of snowmelt as (ii) end of snow cover (EOS) from time-lapse images. EOS, (iii) start-of-season snow-free and (iv) end-of-season snow-covered areas are visualized together as so-called melt layer (Fig. 11 & 12), whereas analysis is conducted separately. SOD/EOD: start/end of snow cover decrease, i.e. last DOY, where SC fraction is at 100 % and first DOY, where SC fraction reaches 0 %, respectively. SOD and EOD are used for the backscatter - SC fraction interaction analysis.

2021; Skov et al., 2019; Westergaard-Nielsen et al., 2017) to assure that the alignment, classification and orthorectification of the time lapse imagery do not induce errors in the seasonal development of SC (Fig. 6a). The correlation of in situ SC fraction

and mean SC fraction of the orthorectified SC fraction maps is tested by calculating the coefficient of determination ($R^2$ = 0.98) and the RMSE (= 3.7 % SC fraction). With such high geolocation accuracy and high agreement in reproducing in situ SC fraction, we can assure minimized systematic errors in the generated SC fraction maps ($SC\_Fraction_{DOY}$). The advantage compared to state-of-the-art validation procedures based on other satellite data is the high spatiotemporal resolution of the generated reference product (2.5 m with a temporal resolution of 1 – 10 days dependent on weather conditions). Thereby, we

can evaluate the response of the SAR signal directly to changes in small-scale SC and the SC fraction.

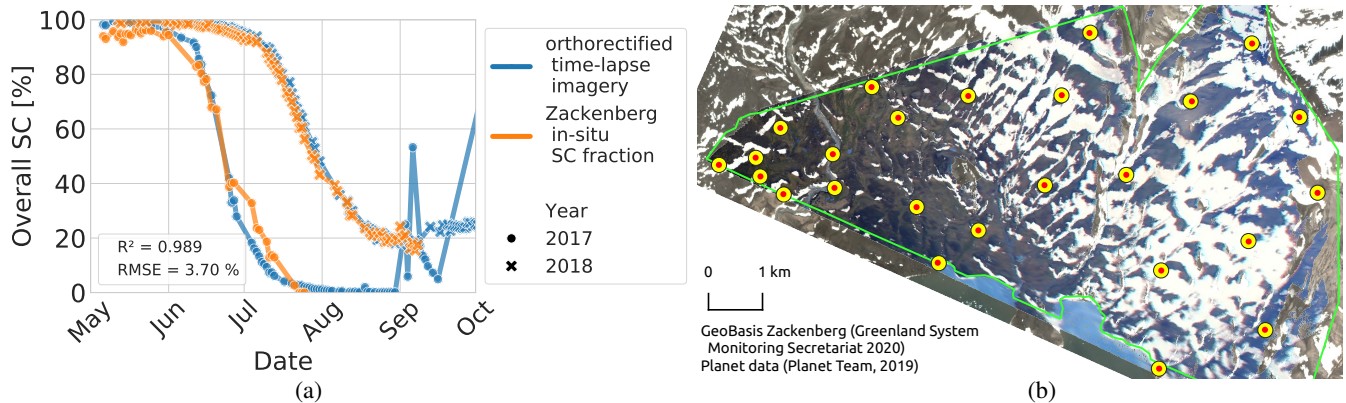

**Figure 6.** (a) Seasonal development of snow cover (SC) deducted from the orthorectified time lapse imagery compared with Zackenberg in situ snow cover fraction data. Their level of correlation is represented by $R^2$ and RMSE. (b) Orthorectified master image and PlanetScope scene in the background with their respective ground control points (GCPs). The centre of the yellow dot indicates the projected location of the GCP, while the red dot shows the true location of the GCP. The distance between both is the orthorectification error (RMSE = 9.4 m).

## 3.3 Evaluation and assessment of the products

### 3.3.1 Backscatter – Snow cover fraction interaction analysis

The development of S-1 backscatter intensity is compared to the SC fraction data derived from the time lapse imagery ($SC\_Fraction_{DOY}$). We use the SC fraction derived from the time lapse imagery as an indicator to examine the timing of backscatter increase. We select all S-1 pixels with sufficient spatial data coverage by the time lapse imagery and include only areas which cover the entire snowmelt development during time-lapse imagery observation. Thus, areas with EOD after 15 August and *start-of-season snow-free* areas are excluded from analysis. Original S-1 $\gamma_0$ intensity data (in dB) as well as intensities, which were rescaled from 0 (melt season minimum $min(\gamma_0)$ intensity) to 1 (mean snow-free summer intensity) are compared to same-day SC fraction data of the time lapse imagery. Furthermore, we identified the temporal distance to SOD and EOD for areas with 100 % and 0 % SC fraction, respectively, in order to capture the development of backscatter intensities before SOD and after EOD. Negative days indicate the number of days before the first observable decrease in SC fraction (SOD). Positive days indicate the number of days after the observed snow cover fraction has reached 0 % (EOD).

### 3.3.2 Assessment of product accuracy dependent on selected threshold and polarization

According to our observations, the selection of polarization and threshold $t$ (Eq. (2)) is crucial for the accuracy of the S-1 snow products of the threshold-based approach. Using the standard threshold (2 to 3 dB) in *Nagler's method* (Nagler and Rott, 2000; Nagler et al., 2016; Snapir et al., 2019) might not be suited due to the different threshold basis (snow-free/dry snow backscatter vs. seasonal minimum) and the use of different levels of preprocessed SAR data ($\sigma^0$ vs. terrain-corrected $\gamma^0$). Therefore, we investigated a threshold range from 2 to 8 dB for HV and from 2 to 10 dB for HH polarization. In addition, we

carried out the same assessment for the derivative approach to compare the results with the best threshold approach. For each
threshold configuration as well as the derivative, the accuracy of the SC maps ($SC\_DOY_{S\text{-}1}$; $SC\_DOY_{camera}$) was assessed
in the following way: The assessment is based only on same-day S-1 acquisitions and orthorectified time lapse imagery to
avoid errors due to temporal offsets (see used dates marked in green in Fig. 3). Acquisitions after 1 September are excluded
from all analyses as our approach using the EOS to derive SC is not capable of detecting new snowfall events in autumn. The
datasets of *start-of-season snow-free* areas and *end-of-season snow-covered* patches are included in the SC accuracy analysis
as follows: The class *start-of-season snow-free* is converted to SC at the start of the season with start-of-season snow-free
pixels assigned as *no-snow*, and not start-of-season snow-free pixels as *snow-covered*. The *end-of-season snow-covered* class
is handled likewise as SC at the end of the season. We calculate confusion matrix metrics (true positive (TP), true negative
(TN), false positive (FP) and false negative (FN) rates as well as overall accuracy) for these SC maps for the selected view-
field of the camera (Fig. 2cd) and derive the receiving operator characteristic (ROC) from the overall TP and FP rate. Further
analysis is carried out for the derivative approach as well as for the global polarization-threshold configuration with the best
ROC (highest TP rate + lowest FP rate). The threshold-based product assessment is considered weaker as it conducted with
the same reference used for identifying this configuration whereas no a priori knowledge is required for the derivative-based
products: (i) The seasonal development of the confusion matrix metrics is assessed. (ii) We compare the EOS layers based on
the areas, where the S-1 product ($EOS\_final_{S\text{-}1}$) and the georeferenced time lapse imagery ($EOS_{camera}$) detect an EOS, by
calculating $R^2$, RMSE and mean absolute error (MAE). (iii) We calculate the difference in EOS DOY between the S-1 product
and the orthorectified time lapse imagery for 2017 and 2018 together and examine the percentage of pixels covered at the same
date as well as within different time ranges of less than 3 ($\pm 2$), 6 ($\pm 5$) and 12 ($\pm 11$) days. The former gives the percentage of
pixels, which were assigned to the temporally closest S-1 acquisition, the latter two correspond to one or two S-1 revisit cycles,
respectively. (iv) Additionally, mean and median of the difference dataset are calculated.

## 4    Results

### 4.1    Interaction between backscatter increase and snow cover fraction

We assess the distinct seasonal backscatter behaviour of S-1 over snow in ZRA and KRA in comparison to the SC fraction
maps based on the orthorectified time lapse imagery ($SC\_Fraction_{DOY}$). For ZRA, the backscatter intensity shows similar
trends in both years and both polarizations. Before the SOD, we observe a longer period of very low backscatter values (20
– 30 days) in 2018, whereas this period is shorter (5 – 10 days) in 2017 (Fig. 7a). The seasonal minimum is reached in ZRA
about 10 to 30 days before the SOD (Fig. 7a). A sharp increase in intensity within the last 10 – 15 days before SOD (Fig. 7ab)
is observed for both years. Neglecting the logarithmic scaling of intensity in the rescaled backscatter intensity, we observe an
increment from about +0.2 above the seasonal minimum to +0.5 to +0.6 above the seasonal minimum before SOD (Fig. 7b).
The remaining increase to 1 (mean snow-free summer backscatter intensity) occurs mostly during the decrease of SC fraction
(Fig. 7bc). The comparison of the absolute HH and HV intensities show differences (Fig. 7a). With HH polarization, a higher
overall intensity and a larger difference between seasonal minimum and mean snow-free intensity is observed compared to HV.

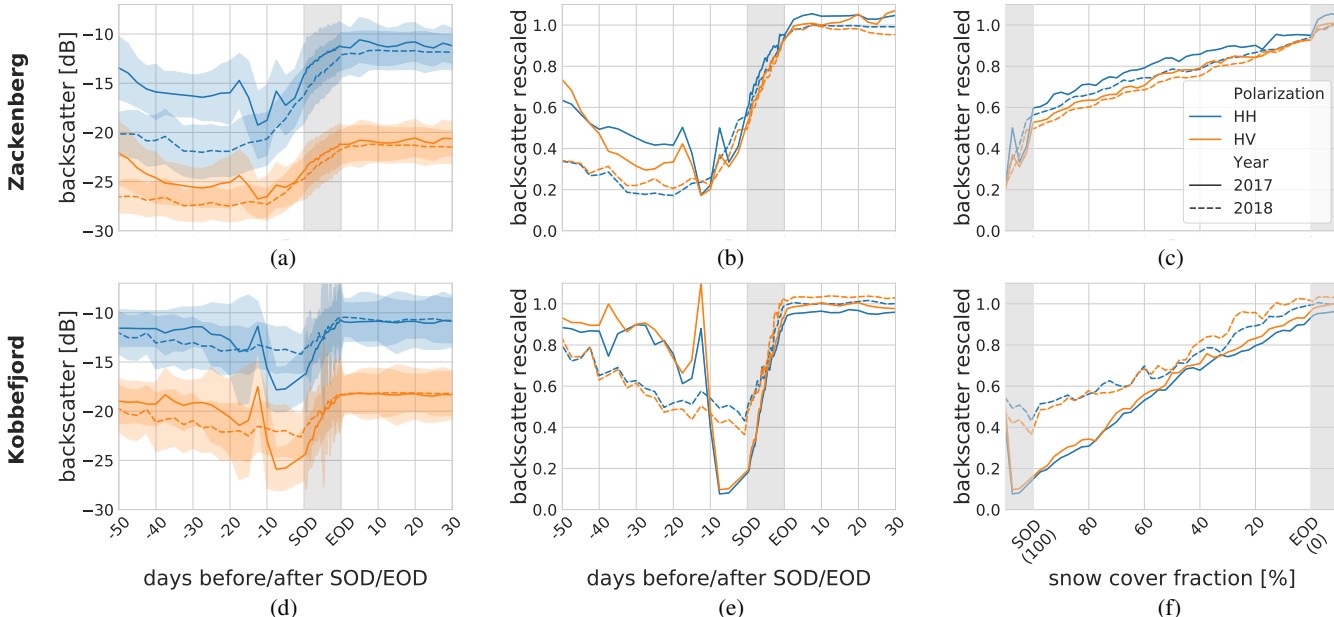

**Figure 7.** Development of Sentinel-1 backscatter intensity over the melting season compared to same-day snow cover (SC) fraction from orthorectified time lapse imagery ($SC\_Fraction_{DOY}$) for ZRA (a-c) and KRA (d-f). Negative days indicate the number of days before the first observable decrease in SC fraction (SOD – start of decrease). Positive days indicate the number of days after the observed snow cover fraction has reached 0 % (EOD – end of decrease). The x-axis range (grey area) between SOD and EOD in (a/d) and (b/e) is modified to the observed average period between these dates (10 days) and does not represent the actual time range. (a/d) Seasonal development of absolute backscatter intensities ($\gamma_0$) with standard deviation and (b/e) of rescaled intensities (0: seasonal backscatter minimum $min(\gamma_0)$; 1: mean snow-free summer backscatter intensity). (c/f) Close view on the development of rescaled backscatter between SOD and EOD during the decrease of SC fraction.

This difference is lower in 2017 than in 2018 due to higher backscatter minima (Fig. 7a). The rescaled intensities are stable for both years and polarizations and we observe a linear increase of rescaled backscatter with increasing SC fraction (Fig. 7c).

In KRA, the period of very low backscatter values is very short and distinct (10 days) in 2017 with a sharp decrease beforehand,
but shows a prolonged period of reduced backscatter intensity for 2018 (Fig. 7e). The seasonal minimum is reached in KRA in less than 10 days before SOD (Fig. 7d). Almost no increase in intensity is observed within the last 10 – 15 days before SOD (Fig. 7de) for both years unlike in ZRA. The increase in the rescaled backscatter intensity shows again a linear behaviour and occurs almost entirely during the decrease of SC fraction in 2017 and 2018 only that in 2018 the averaged backscatter value is not reaching values lower than 0.4 (Fig. 7ef). The absolute backscatter values indicate generally a lower seasonal
variability in KRA than in ZRA, especially for 2018 due to higher backscatter minima. The snow-free HV backscatter intensity in Kobbefjord is generally higher than in Zackenberg, whereas no such difference is visible in HH (Fig. 7d).

## 4.2 Selection of threshold, derivative and polarization for Sentinel-1 snow products

We assessed the seasonal SC development for each threshold $t$ in combination with each polarization to identify the parameter configuration that best fits the seasonal SC development in the orthorectified time lapse imagery (Fig. 8, 9 & 10). We further compared these results with the derivative based approach.

For the determination of $t$, two opposite developments influence the accuracy of the resulting datasets in ZRA: The higher $t$, the more areas are mistakenly classified as *start-of-season snow-free* (Fig. 10a), as intensities of these locations do not exceed the defined threshold. In contrast, EOS is detected better using higher $t$, whereas low $t$ values cause a negative offset and EOS is detected earlier than observed (Fig. 10b). The constant increase of the intensity during the melt period causes this earlier detection of EOS with lower $t$ values. Thereby, higher values of $t$ result in an underestimation of SC in the early melt period, whereas lower values of $t$ lead to an underestimation of SC during the later melt period (Fig. 8). In KRA a similar development for overestimation of *start-of-season snow-free* areas is detected (Fig. 10c), but the melt offset is generally moved towards positive values shifting the optimum towards lower $t$ (Fig. 10d). Thereby, the highest SC accuracies are reached with lower thresholds independently of the used polarization. Comparing the two polarizations, we observe a higher negative offset in EOS detection for HH (Fig. 10b), but slightly lower overestimation of *start-of-season snow-free* areas (Fig. 10a) in Zackenberg, and the opposite trend for Kobbefjord (Fig. 10cd). The derivative approach adapts to differences in optimal thresholds (especially for ZRA in HH), resulting in a reduced overestimation of *start-of-season snow-free* areas for ZRA and KRA, while offset in EOS detection remains low ($\pm$ 3 days) for ZRA (Fig. 10b) and stable around +5 to +7 days for KRA (Fig. 10d). New SC due to snowfall events in autumn is not detectable by the proposed approaches (Fig. 8 & 9).

As the ROC analysis in Fig. 8cf and Fig. 9cf indicates, the $t$ values for the best results depend on the used polarization and the observed year and site. Higher TP rates in 2018 compared to 2017 are observed for both sites (Fig. 8cf & 9cf). In ZRA, the optimal value for $t$ differs between the years with higher values found in 2018 (e.g. for HV from 3 to 4 dB in 2017 to 4 to 5 dB in 2018; Fig. 8cf). The increase of the optimal threshold is higher in HH than in HV (Fig. 8cf). Hence in Zackenberg, the performance of a global threshold is more robust for HV. In Kobbefjord, the ROC shows no large differences between the years. The best accuracies are reached for both polarizations with a threshold of 2 to 3 dB. HV tends to produce higher TP rates than HH in Kobbefjord (Fig. 9cf). In Zackenberg, the derivative approach works equally well for both polarizations and the difference to the optimum threshold configuration is low. In Kobbefjord, the derivative approach using the HV polarization outperforms HH. The derivative generates lower accuracies in 2018 whereas, in 2017, the reached accuracies are equal to the optimum threshold. We use the global threshold configuration that shows the best ROC response for both sites and years (HV – 4 dB) as well as the derivative based on HV for further analysis. The analysis using the threshold-based products is considered weaker than the derivative method, as a priori knowledge is required to identify the optimal global threshold. The resulting layer composites of EOS, *start-of-season snow-free* and *end-of-season snow-covered* for this configuration are shown in Fig. 11 for ZRA and in Fig. 12 for KRA.

The seasonal development of confusion matrix parameters (Fig. 13, two left columns) shows that overall accuracy is always above 75 % and in more than half of the cases above 90 %. The lowest overall accuracy occurs during the melt period, when

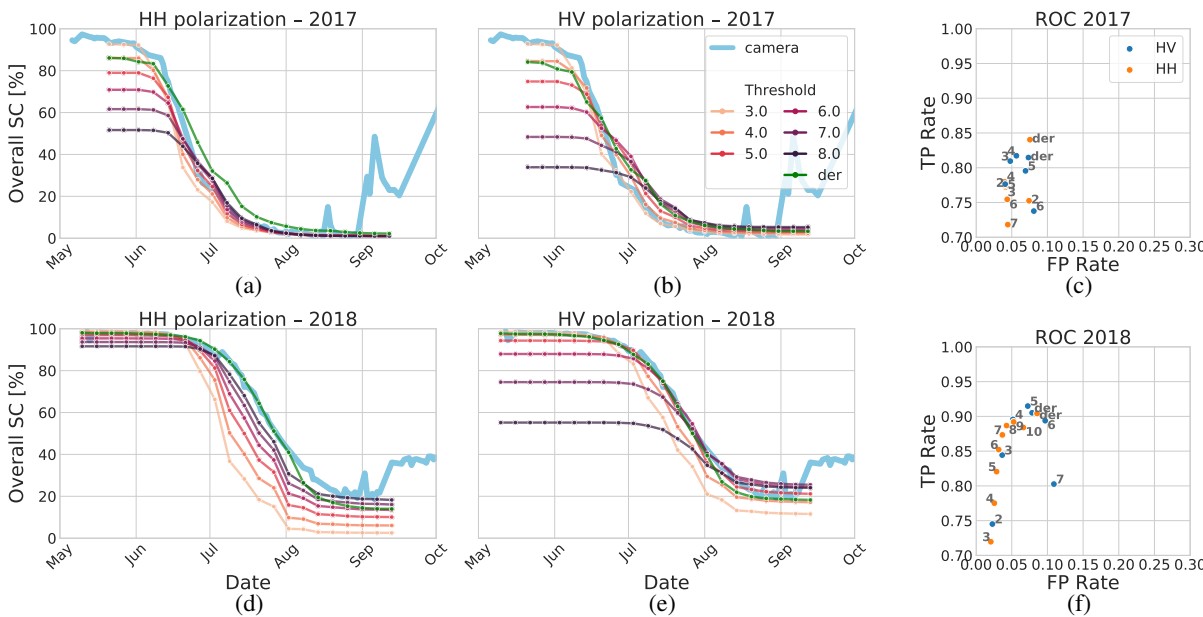

**Figure 8.** (a,b,d,e) Seasonal development of overall snow cover (SC) in Zackenberg of the Sentinel-1 products ($SC\_DOY_{S-1}$) for the threshold-polarization configurations as well as the derivative (der) compared to orthorectified time lapse imagery ($SC\_DOY_{camera}$). Autumn SC is not detected by the proposed approach as only the seasonal decrease of SC is observable. (c,f) Receiver operator characteristic (ROC) derived from the overall true positive (TP) rates and false positive (FP) rates of the SC maps of (c) 2017 and (f) 2018 based on same-day S-1 acquisitions and orthorectified time lapse imagery. The used threshold is depicted as label next to each point.

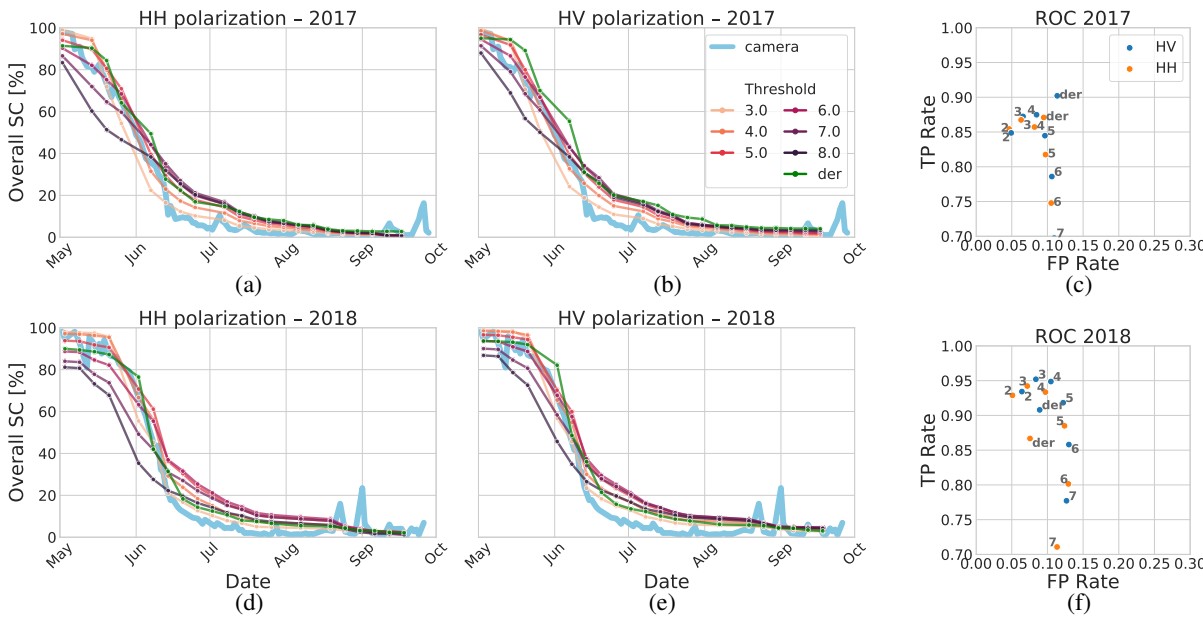

**Figure 9.** (a,b,d,e) Seasonal development of overall snow cover and (c,f) receiver operator characteristic (ROC) of Kobbefjord.

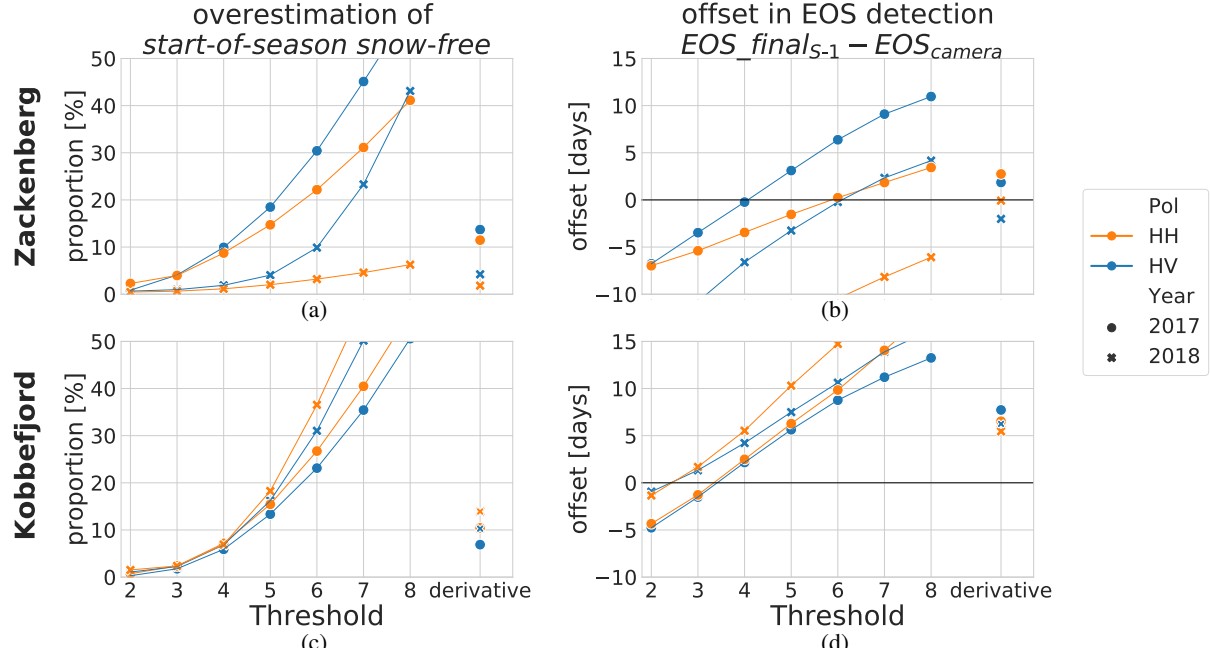

**Figure 10.** The key snow cover (SC) mapping accuracy parameters influenced by the threshold and polarization setting are (a/c) overestimation of start-of-season snow-free areas for higher thresholds leading to an underestimation of SC in early melt season and (b/d) a negative offset in mean EOS for low thresholds leading to underestimation of SC in late melt season.

melt is the strongest and the decrease in SC is the highest. In ZRA, FP and FN are occurring during early melt season in 2017, whereas in 2018 mostly FN during late melt are observed. The comparably high proportion of FP responses (14.6 %) in 2017 on DOY 177 (Fig. 13ae) is caused by a late snowfall event in about the beginning of July undetected in the S-1 dataset. As we observe predominantly FN almost throughout all observations, SC is generally rather underestimated by the S-1 product

in ZRA. An underestimation of early-season SC induced by an overestimation of *start-of-season snow-free* areas (Fig. 10ae) is observed in 2017, whereas in 2018 the observed underestimation of late-season SC (Fig. 10bf) as well as *end-of-season snow-covered* is caused by the temporal offset in EOS detection (Fig. 10b). This indicates that a global threshold of 4 dB might be slightly too high for 2017 and slightly too low for 2018 in ZRA, which is consistent with the ROC analysis (Fig. 8cf). In KRA, predominantly FP is observed during the melting period resulting in a slight overestimation of SC by the S-1 product

(Fig. 13).

For the EOS product, a $R^2$ score of 0.41 (p<0.001), a RMSE of 13.5 days and a MAE of 9.4 days is observed in ZRA with the global threshold. Using the derivative increases the accuracy to an $R^2$ score of 0.63 (p<0.001), a RMSE of 11.6 days and a MAE of 8.1 days. The observed accuracy measures in KRA are low ($R^2 < 0$) due to the limited variation of EOS in the camera field of view (Fig. 12). The regression density plots in Fig. 13 show the correlation and the histogram plots in Fig. 13 show the

375 range of temporal difference in days between the two datasets. Within $\pm 2$ days, $\pm 5$ days and $\pm 11$ days, 15 % to 26 %, 46 % to 49 % and 72 % to 80 % of all pixel-wise EOS dates are detected for both sites and both approaches. Using the threshold, the

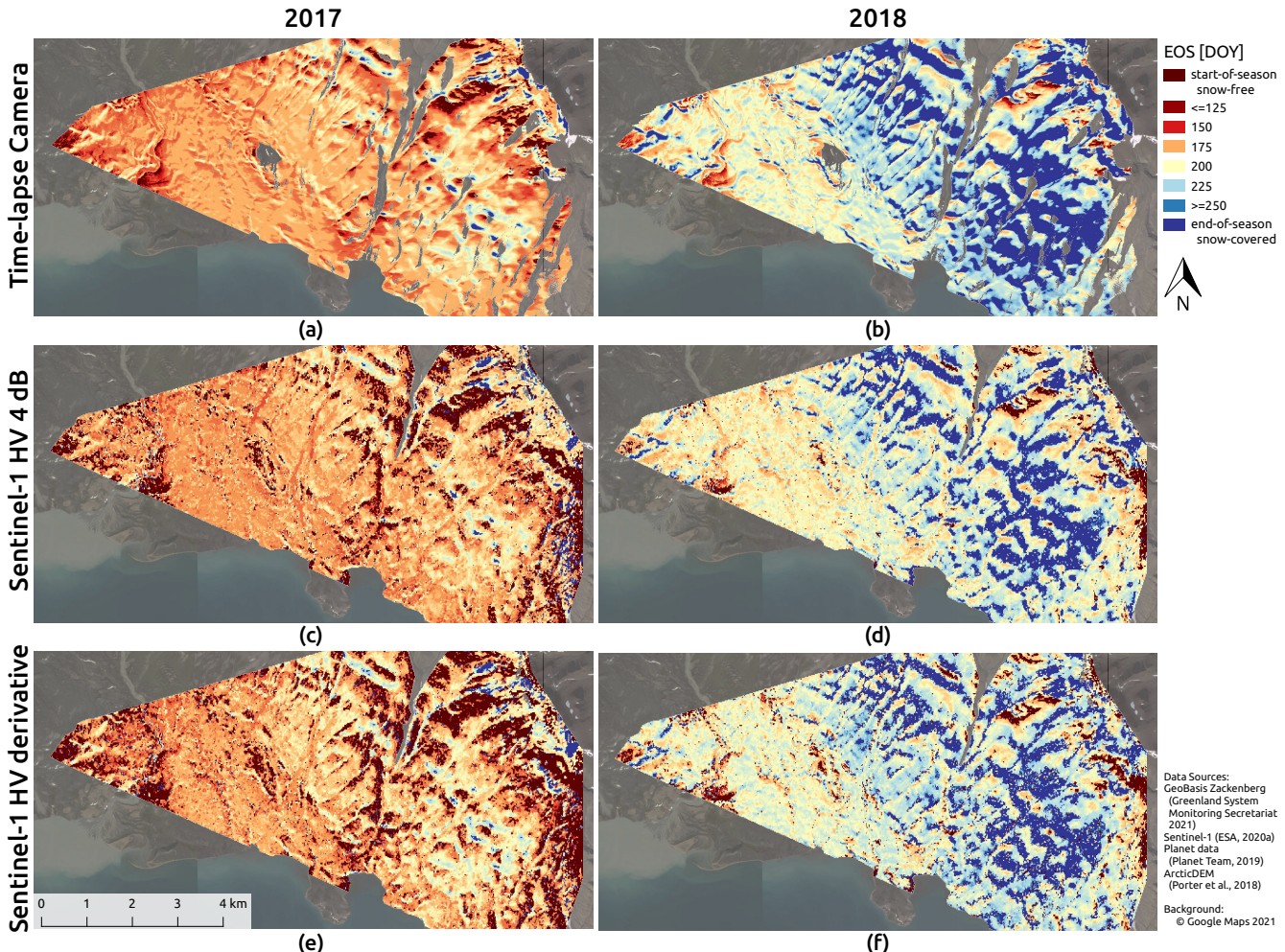

**Figure 11.** Melt layers (composites of day of year (DOY) for end of snow cover (EOS), *start-of-season snow-free* areas and *end-of-season snow-covered* patches) of Zackenberg: (a-b) Reference product by the time-lapse imagery with 2.5 m resolution for (a) 2017 and (b) 2018; Subset of the generated S-1 product with 20 m resolution bounded by the camera field of view with global parameter configuration (polarization: HV; threshold $t$: 4 dB) for (c) 2017 and (d) 2018; (e-f) the same area of the S-1 product with the derivative approach for (e) 2017 and (f) 2018.

mean (-3.1 days) and the median value (-4 days) of the histogram are slightly negative in ZRA whereas, with the derivative in ZRA, mean (0.0 days) and median (+1 days) are close to zero. In KRA, the threshold-based values of mean (+3.1 days) and median (-1 days) are slightly higher than in ZRA. Using the derivative approach shows, like in ZRA, a trend towards higher values (mean: +6.8 days; median: +3 days). This is consistent with the observed offsets in EOS detection (Fig. 10bd) and the distribution of EOS shown in Fig. 11 & 12.

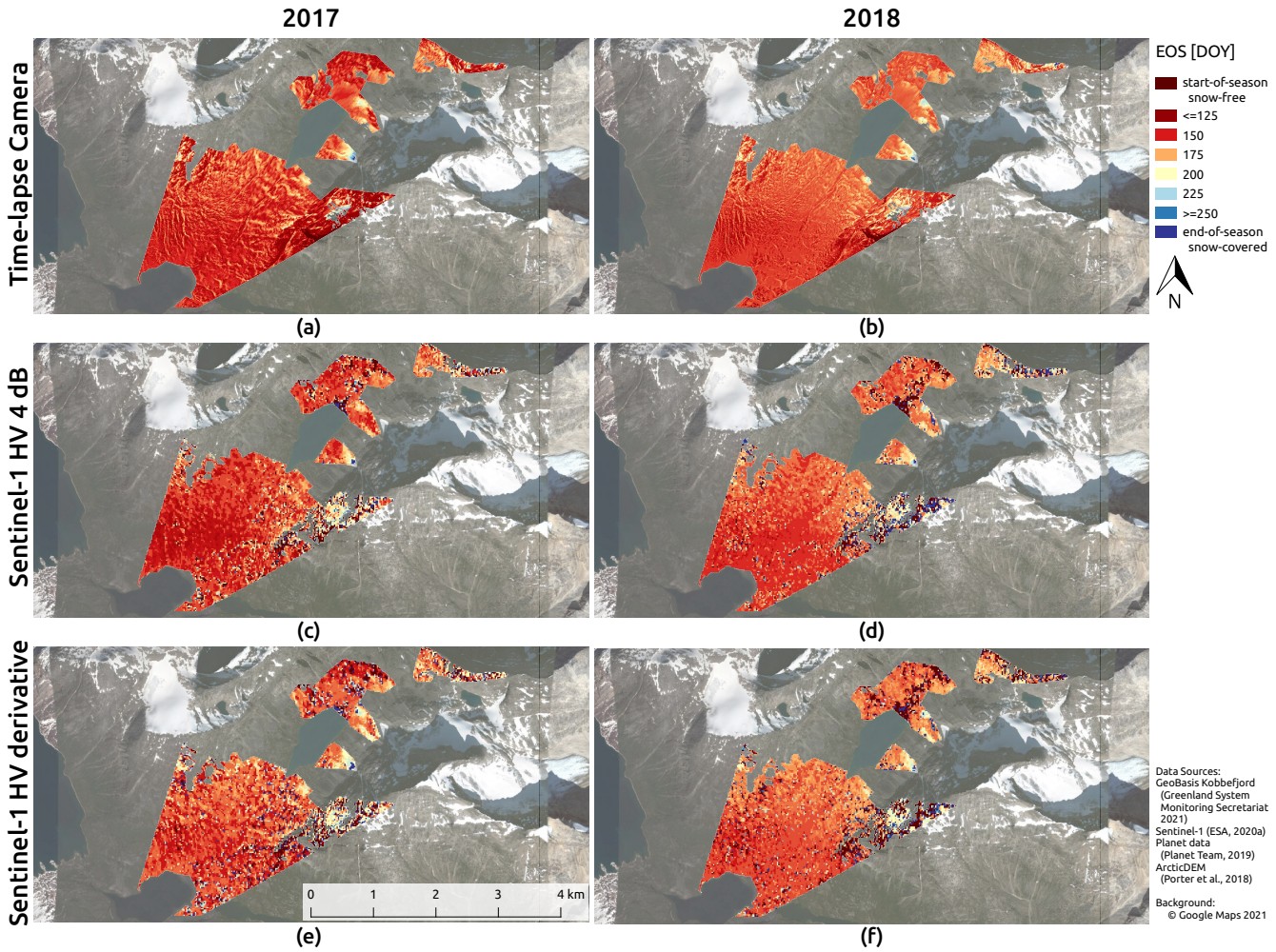

**Figure 12.** Melt layers of Kobbefjord: (a-b) reference product by the time-lapse imagery reference; (c-d) subset of the S-1 product with global parameter configuration and (e-f) with the derivative approach.

## 5 Discussion

### 5.1 Influence of SC fraction and snow properties on SAR backscatter intensity

We observe a distinct seasonal behaviour in S-1 C-band backscatter with a clear decrease during early melt, reaching a mini-
385 mum just prior to the onset of melting, followed by a constant linear increase towards the EOS. This is consistent with findings in Marin et al. (2020). According to our observations, this development, which is found in both polarizations (Fig. 7a), is probably driven by changes of the snowpack, e.g. increased surface roughness, larger size and number of snow grains like suggested by Marin et al. (2020), as well as by decreasing fractional SC. The prolonged period of low backscatter values and the higher variance in absolute backscatter intensity in ZRA compared to KRA is probably caused by overall extended melting periods

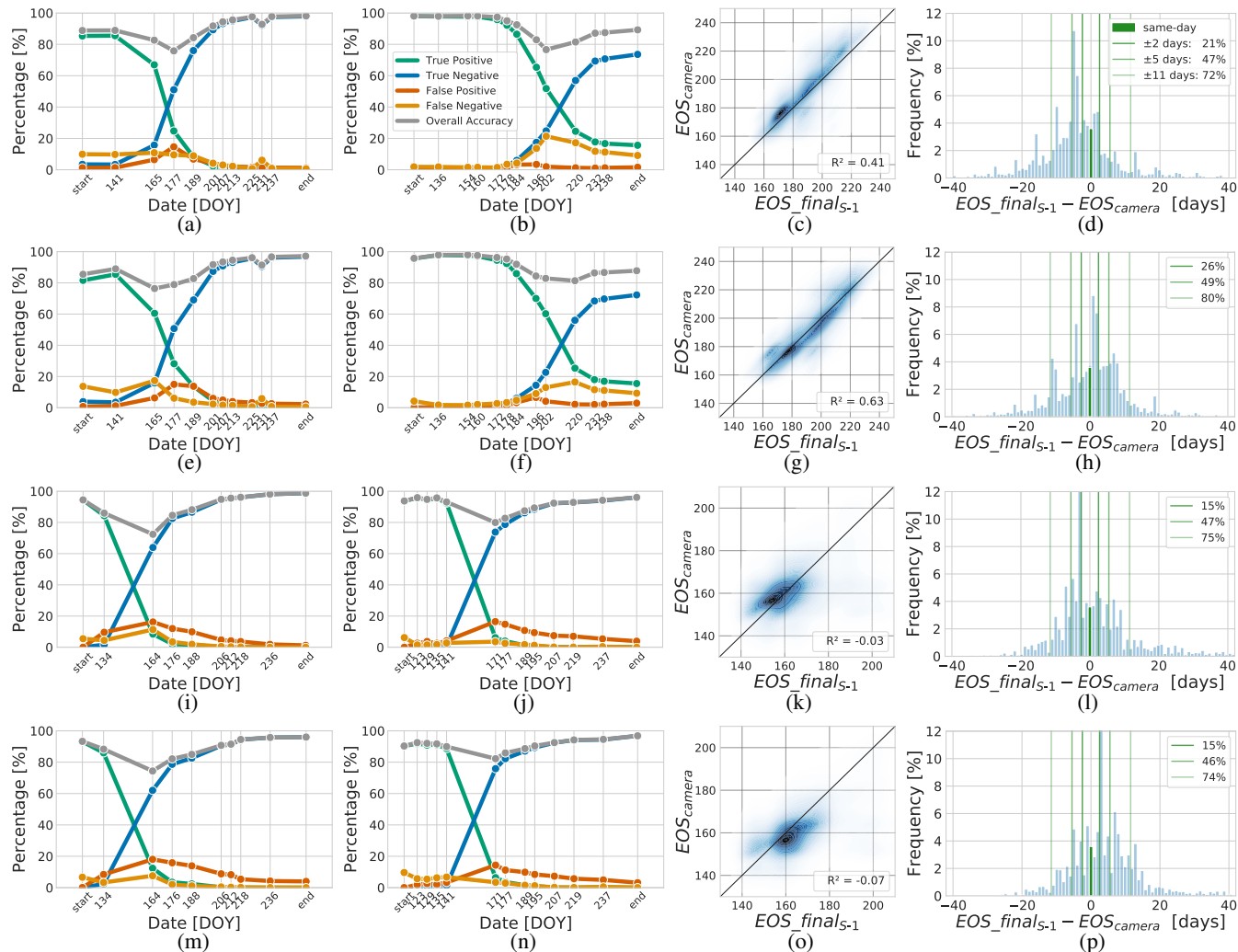

**Figure 13.** (a,b) Seasonal development of true positive, false positive, true negative and false negative as well as overall accuracy of the snow cover maps of the year (a) 2017 and (b) 2018. Overall accuracy is always above 75 % and in more than half of the cases above 90 %. Generally, false negative responses or underestimation of snow cover is dominant. False positive responses or overestimation occurs mostly in 2017 around DOY 177. (c) Correlation between EOS detection from S-1 time series approach and time lapse imagery. (d) distribution of temporal offsets in EOS. Highlighted are used time ranges corresponding to 0.5, 1 and 2 Sentinel-1 revisits. (a-d) ZRA 4 dB threshold. (e-h) ZRA derivative. (i-l) KRA 4 dB threshold. (m-p) KRA derivative.

in ZRA and possibly also influenced by the different acquisition times in afternoon and morning, respectively. The afternoon acquisition usually detects wet snow earlier in the melt season due to the diurnal melt and refreeze cycle of the snowpack. In ZRA, about half of the increase in intensity occurs within 10-15 days before the decrease in SC fraction starts (Fig. 7ab), whereas no such development is observed in KRA (Fig. 7de). The increase before SOD in HH is probably caused by the higher

surface roughness, while increased depolarization at this rougher snow surface increases the backscatter intensity in HV. The absence of a pre-SOD increase in KRA could be either due to the different acquisition time in the morning, but is more likely caused by systematic differences in SC distribution between ZRA and KRA: (i) snow depletion is occurring faster in KRA than in ZRA leaving less time for alteration of the snowpack before depletion. (ii) snow distribution shows higher heterogeneity on small scales resulting in smaller snow patches in KRA in comparison to ZRA. Hence, subpixel parts are turning snow-free almost directly after start of melt is observed. Another sign for this systematic difference indicating more homogeneous snow distribution on pixel scale in ZRA compared to KRA is the temporal difference between SOD and EOD, which is about 10 days in ZRA, but more than 20 days in KRA. The remaining intensity increase in ZRA and almost all increase in KRA occurs along with decreasing SC fraction and is almost linear. It is potentially driven by the increasingly higher proportion of the signal coming from snow-free parts of the pixel and possibly a further alteration of the snowpack (Fig. 7c). This linear increase seems suited to derive SC fraction from SAR backscatter intensity (e.g. Luojus et al. (2006) and Koskinen et al. (2009)); however, such an approach would need to address the varying strength of pre-SOD increase in intensity and effects from changing surface properties underneath the snow, speckle and the viewing geometry, which result in variability of the SAR signal and make the discrimination between snow-free areas and areas with patchy wet snow in threshold-based SAR approaches like *Nagler's method* (Nagler et al., 2016; Nagler and Rott, 2000), as well as in our approach, challenging. These results are of comparably high validity, as the observations have been compared to a high-resolution reference dataset of same-day SC fraction maps derived from orthorectified time lapse imagery.

## 5.2 Influence of threshold and polarization on the products

For the selection of thresholds, we observe two main drivers for SC mapping inaccuracies: (i) increased underestimation of SC during early melt linked to an increased overestimation of *start-of-season snow-free* areas with higher $t$ (Fig. 8, 9 & 10ac); (ii) increased underestimation of SC during late melt linked to an increased offset of earlier EOS detection with lower $t$ in ZRA (Fig. 8 & 10b). For an accurate result, these contrary effects need to be balanced. While using a season-independent global threshold leads to a better performance in HV compared to HH due to the lower absolute seasonal changes in backscatter intensities in HV (Fig. 7a), season-dependent thresholds can produce accurate results in both polarizations, but require in situ reference data. The better global performance of cross-polarization for SC detection is in accordance with other studies applying *Nagler's method* (Nagler et al., 2016; Thakur et al., 2018), which also indicated better performance of the cross-polarization compared to the co-polarization channel. The lower ROC performance in 2017 compared to 2018 (Fig. 8cf) could be caused by the limited length of the time series in ZRA and gaps in the time series in KRA. The increase towards higher values of the best fitting $t$ for ZRA in 2018, which is in accordance with the observed higher seasonal backscatter difference (Fig. 7a), is possibly caused by higher overall snow depths observed in 2018 by López-Blanco et al. (2020). The slightly lower optimal thresholds (2 to 3 dB) in KRA might be caused by the lower variance in backscatter intensity due to the different acquisition time. The seasonal defined threshold can vary about $\pm 1$ dB around the optimum while still giving good results (Fig. 8cf), which indicates that the used global threshold in HV for 4 dB is applicable. The derivative-based approach generates similar results as the optimal seasonal threshold. However, no optimization or further in situ reference data is required. The

systematic positive offset in EOS detection for KRA shows a later detection of EOS by S-1 than compared to the camera (Fig. 10d). Hence, the derivative-based EOS might be sensitive to lower SC fraction than defined by $EOS_{camera}$ (SC fraction < 50 %) and possibly also detects low fractional SC. This could also be the cause for the increased FP rates of the derivative approach compared to the optimal seasonal threshold (Fig. 8cf & 9cf). The lower TP rates of KRA in 2018 (Fig. 9f) might be caused by an increased overestimation of *start-of-season snow-free* areas.

The analysis using the threshold-based products is considered weaker than the derivative method, as a priori knowledge is required to identify the optimal global threshold. However, the degree of optimization for the threshold setting is reduced to a minimum with a global threshold instead of training a season and site dependent threshold. The approach using the derivative is less susceptible to this issue.

Using HV and $t = 4$ dB , EOS is detected with a reasonable accuracy within two S-1 observations (Fig. 13). Potentially a denser time series incorporating different S-1 orbits could improve the accuracy in snowmelt detection; however, different orbits need to be analyzed separately, due to differences in local incidence angle and, more importantly, different acquisition times, which need to be considered. However, our case study shows that our approach works well with both settings as different orbits (ascending in ZRA & descending in KRA) and, hence, different acquisition times (afternoon and morning, respectively) were used. The SC maps reproduce the overall SC development with nine out of ten SC maps above 80 % overall accuracy and more than half above 90 % (Fig. 13) while most of the error is due to underestimation of SC in ZRA and due to overestimation in KRA. Thereby, accuracies comparable to other latest SC mapping approaches using optical remote sensing with similar spatial resolution (e.g. Gascoin et al., 2019; Girona-Mata et al., 2019; Piazzi et al., 2019) are generated but with the advantage to be independent from cloud cover. However, it has to be pointed out that autumn and episodic SC due to snowfall events are not detectable by the proposed approach as only the seasonal depletion of SC during melt is observable. Further, dense vegetation in other study areas might cause increased inaccuracies due to the insensitivity of C-band SAR for snow in such areas (Nagler et al., 2016; Tsai et al., 2019c).

The differences between the threshold- and derivative-based approach are rather low, which indicates that both might be applicable on other sites. However, derivative has the advantage to automatically adapt to different seasonal and environmental conditions. A systematic difference is observed in the detection of EOS indicating a higher sensitivity of the derivative for low fractional SC.

## 5.3   Major advantages of the proposed approach compared to other recent SAR based snow cover studies

With this new approach multiple advances are made compared to other recent studies on SAR-based SC detection and the current standard, *Nagler's method*: (i) we use the entire time series instead of only a few images per year unlike most previous studies (according to Tsai et al., 2019b); (ii) we avoid the manual selection of reference images for *Nagler's method* (Nagler and Rott, 2000) and omit challenges like finding a completely snow-free or dry snow scene as well as a potential deterioration of the reference caused by altered backscatter signals due to end-of-season SC and firn. Due to the simple backscatter threshold and derivative approaches, we keep analysis fast and reduce processing capacity compared to the supervised classification approach by Tsai et al. (2019a, c), who additionally calculated interferometric and polarimetric features. Even though the spatial

resolution is lower in their approach (100 m), resulting overall accuracies for similar low vegetated areas are comparable to the ones observed here (Tsai et al., 2019c). Threshold setting must be assessed in more detail to confirm, whether a global threshold is applicable also in other sites and years, and to analyse which effects different snow properties, vegetation, substrate and local incidence angle might have on the seasonal backscatter behaviour of S-1 above snow, e.g. seasonal minimum & variation, as well as the resulting product accuracy. The derivative adapts well to different environmental settings (high & low arctic) and different seasonal conditions (high and low snow depths). Hence, the derivative approach could be applied to another site without requiring prior optimization of the threshold. Further, both approaches are based on the physical principle of SAR backscatter during snowmelt. Hence, they are expected to work well in other low vegetated areas as the characteristic seasonal pattern has been observed in the Alps (Marin et al., 2020) as well as around the globe (Lievens et al., 2019). Parametrisation, e.g. of the threshold or the melt season, however, must be adapted and adjusted to local conditions. Moreover, further research needs to address the transferability to areas with denser vegetation or human induced activities which could influence the SAR backscatter signal.

With this new approach, many relevant parameters for SC monitoring are detected at a weekly basis by the here proposed approach: State and extent of SC during melt, end-of-season SC and start-of-season snow-free areas. Further, important hydrological measures like start of runoff (SOR) and end of snow cover (EOS) are derived, whereas the former is not detectable with optical remote sensing data. Potentially, the snow phase detection algorithm by Marin et al. (2020) could be incorporated to further separate melt phases in more detail and *Nagler's method* (Nagler and Rott, 2000) could be used to identify the start of wet-snow phase. Provided at a spatial resolution of 20 m, hydrological models could further use this information to derive additional parameters like snow water equivalent (based on reconstruction approaches presented e.g. by Molotch and Margulis (2008); Kerr et al. (2013); Bair et al. (2016); Rittger et al. (2016)) or assess the delay of snowmelt runoff due to melt water storage in the snowpack and the soil (Marin et al., 2020; Tsai et al., 2020). If available, different S-1 orbits could be used to increase the temporal resolution of the product, but need to be analyzed separately due to differences in local incidence angle and acquisition times. Further, EW data could be used with our approach for snowmelt detection and snow cover depletion mapping on larger scales with coarser resolution. Thereby, our approach enhances monitoring of hydrological cascading effects and could support in combination with other methods and datasets a holistic hydrological monitoring of SC from the scale of a single catchment up to pan-Arctic observations.

## 6 Conclusions

In this study, we present a fast and simple approach for mapping snow cover (SC) and timing of snowmelt based on Sentinel-1 (S-1) synthetic aperture radar (SAR) time series. Using the distinct seasonal signal of backscatter intensity above snow, the approach employs user-defined thresholds based on the seasonal backscatter minimum as well as the derivative of the time series to (i) identify start of runoff (SOR) and end of snow cover (EOS) as day-of-year (DOY), (ii) detect start-of-season snow-free areas and end-of-season snow-covered patches and (iii) derive a SC extent map for each S-1 observation date during SC

depletion. EOS and SC are compared to maps derived from aligned and orthorectified terrestrial time lapse imagery providing much higher spatial (2.5 m) and higher temporal (1 to 10 days) resolution than the S-1 product.

We compared the seasonal evolution of the SAR backscatter intensity to orthorectified SC fraction maps based on same-day time lapse imagery. We observe that in ZRA about half of the HH and HV backscatter intensity increase during snowmelt occurs within 10-15 days before the decrease in SC fraction starts, whereas no such increase was observed in KRA due to faster melt and, hence, less time available for snowpack alteration. From then onwards, backscatter increases linearly with decreasing SC fraction. Hence, changes in the snowpack (e.g. grain size and number, surface roughness) as well as decrease of fractional SC are drivers for the observed backscatter increase.

The new approach to map SC and snowmelt was tested with HH and HV polarizations and different backscatter intensity thresholds (2 to 8 dB) indicating the following major error sources: (i) underestimation of SC during early melt due to an over-estimation of start-of-season snow-free areas caused by parts which do not exceed the selected backscatter threshold (increasing with higher thresholds); (ii) underestimation of SC during late melt as well as end-of-season SC due to a systematically earlier detection of EOS (increasing with lower thresholds) in ZRA; (iii) neglection of episodic and autumn SC due to snowfall events as only the depletion of SC is detectable. The variation in the optimum threshold is higher in HH, which causes HV to produce better results with a global threshold. Using a global threshold of 4 dB with the HV polarization, EOS is correctly assigned to the closest S-1 acquisition for 15 to 27% of the area, while 45 to 49 % and 72 to 80 % are correctly detected within a period of one and two S-1 repeat cycles (6 and 12 days). The resulting SC maps are generated with an overall accuracy of always more than 75 % and in more than half of the cases above 90 %. Using the derivative instead produced similar results and adapts well to different environmental settings and seasonal conditions. Hence, the derivative approach could be applied to another site without requiring prior optimization of the threshold. Both approaches work well with different orbits and acquisition times. Being based on the physical principle of SAR backscatter during snowmelt and its characteristic seasonal pattern, the approach is expected to work well in other low vegetated areas around the globe, but further research is required to confirm the transferability of the approach to other settings and especially also areas with denser vegetation or human induced activities which could influence the SAR backscatter signal. Finally, a SC product with this spatiotemporal resolution (20 m – 6 days) is, to the best of our knowledge, not presented with any other open-data remote sensing approach.

Further improvement could take advantage of the combined use of different S-1 orbits to increase the temporal resolution of the product or incorporate the snow phase detection algorithm by Marin et al. (2020) and use *Nagler's method* (Nagler and Rott, 2000) to detect the start of the wet-snow phase. Thereby, continuous S-1 snow monitoring could not only improve hydrological and climatological models but also lead to an enhanced understanding of the complex interactions between climate change, SC and the arctic ecosystem.

*Code and data availability*. Relevant code and data can be made available upon request to SB. All Sentinel-1 data are freely available at https://search.asf.alaska.edu/ (Alaska Satellite Facility, 2020) and https://scihub.copernicus.eu/ (ESA, 2020a) upon registration. In situ

data from Zackenberg and Kobbefjord research area are freely available upon registration at https://data.g-e-m.dk/ (Greenland Ecosystem Monitoring Secretariat, 2020).

*Author contributions.* SB conceptualized this study, carried out analysis (methodology, visualization) and wrote the paper. KS and KR assisted with access to GeoBasis data. TU provided resources and supervision. All co-authors assisted during the writing process and critically discussed the content.

*Competing interests.* The authors declare that they have no conflict of interest.

*Acknowledgements.* Time lapse imagery and snow cover fraction data from the Greenland Ecosystem Monitoring Programme were provided by the Department of Bioscience, Aarhus University, Denmark in collaboration with Department of Geosciences and Natural Resource Management, Copenhagen University, Denmark and Asiaq Greenland Survey, Greenland, Denmark. The Greenlandic Research Council NIS is acknowledged for project support. We thank the Education and Research Program by Planet Lab (Planet Team, 2020) for providing free access to PlanetScope satellite imagery for scientific purposes; and the Polar Geospatial Center for providing free access to the ArcticDEM under NSF-OPP awards 1043681, 1559691 and 1542736. We gratefully thank the four anonymous reviewers for their valuable comments.

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
