# Peer review of "Sentinel-1 time series for mapping snow cover depletion and timing of snowmelt in Arctic periglacial environments: Case study from Zackenberg and Kobbefjord, Greenland"

_The Cryosphere, 2021_

## Author Response (AR1)

**General remarks on the revision and the reviewer's comments:**

Dear reviewers,

the authors appreciate your comments, and we thank you for your valuable work and the provided suggestions and comments on our contribution. We have addressed all your remarks and a point-to-point response (authors responses are in italic) to your questions/comments is provided below. The revised version of the manuscript is ready for submission, once requested.

From your reports we have identified the following major issues. Before the point-to-point response, we like to point out how these were addressed:

(1) Use of one orbit and potential use of EW?

An explicit goal of our investigation is to provide small-scale estimates of SC and related parameters and to study the temporal evolution of the SAR signal in relation to high-resolution in situ data, which provide snow cover fraction estimates. As such we have not considered the use of EW data as these (i) will not allow to study the small-scale SC heterogeneity due to their coarse spatial resolution and (ii) are not suited for the rather small sizes of the test sites (42 and 7 km² covered by the in situ cameras). We have further just used one relative orbit (IW), as acquisition geometry needs to be constant throughout the time series to ensure a comparability of the measurements. Nevertheless, we point out in the discussion that additional orbits (IW), if available, might be used to densen the time series; however, different orbits need to be analyzed separately, due to differences in local incidence angle and acquisition time. Nevertheless, we believe that EW data could be used with our approach for snowmelt detection and snow cover depletion mapping on larger scales with coarser resolution in future studies.

(2) Decrease in spatial resolution might lead to better results?

The decrease in spatial resolution might in fact lead to better results, as a generalization will cause a better signal to noise ratio (reduction of variance) and a better radiometric stability (less speckle noise when increasing the number of looks). However and similar to the first answer, we wanted to make use of the high-resolution in situ camera imagery and the Sentinel-1 IW data. As such, we explore if it is possible to estimate SC and related parameters on comparable small-scale using S-1 time series and to infer effects related to the SC fraction cover during the melt, i.e. a high-spatial resolution is inherently required to observe/characterize this processes due to the patchiness of SC during the depletion.

(3) Limited transferability → use derivative instead of fixed thresholds

Thanks for your suggestions on this. We have now included an approach that uses the derivatives of the time series and, therefore, operates more adaptively. It is presented along with the threshold-based approach. Results point out that accuracies similar to the ones achieved from the threshold-based method can be realized. As now discussed, an approach using

derivatives is believed to be less sensitive to signal-differences caused by different conditions of the snowpack or the land cover. Therefore, the derivative-approach favors transferability.

(4) Limited transferability → add another site to test the capabilities

We have included a second test site and now show results also for the Kobbefjord region (Western-Greenland close to Nuuk). The Kobbefjord research area is, like Zackenberg, part of the Greenland Ecosystem Monitoring programme. Therefore, it offers a similar setup and also time-lapse camera imagery of the valley is available. As indicated in the revised version, we have repeated the entire processing of the camera imagery and of the Sentinel-1 time series for the Kobbefjord test site and we present results of both regions. Note that the environmental setting in Kobbefjord (low Arctic) is different to the setting in Zackenberg (high Arctic), which is also evident when studying the SC and its temporal evolution. Even though, the presented methods (threshold- and derivative-based approaches) perform well for both sites and produce reliable estimates, which compare well with the in situ measurements. For sure this is not a proof for a truly "global applicability" (which is also outside the scope of the contribution), but results confirm that the general design of the approach is not over-fitted but transferable.

(5) Factors influencing the threshold setting (vegetation, snow depth, soil properties)

It is correct that factors influencing the setting for the threshold-based approach cannot fully be captured by the reference data available, as such their influence on the threshold setting itself cannot be studied, nor is it possible to explain the influence on the S-1 signal in detail. From the time lapse imagery, only the influence of the SC cover fraction on the backscatter can be compared and analysed, while information on snowpack properties is missing. This issue is now better addressed in the discussion and also points to future research needs. Note as well in this context that the derivative-approach favors transferability as it is self-adjusting and not linked to a fixed global (scene) threshold.

(6) Terminology and Abbreviations

According to your recommendations, we had a look at all terms used in our manuscript and redefined them to fit better along with terms used by other publications. We like to thank Reviewer 4 for the recommendation to use Fig. 1 for that. We added the parameters there and present the adapted graphic below. Besides, a table with all changed terms is shown below:

[Figure]

| Old term | New term | Explanation |
|---|---|---|
| Start of snowmelt (SOS) | Start of runoff (SOR) (solely detectable by S-1) | Reviewer 3 correctly commented that this approach using the backscatter minimum as so-called start of snowmelt is actually not in line with other publications and the term might be ambiguous due to the different phases of snowmelt, i.e. SOS could also be at the beginning of the moistening phase / wet snow phase. As we use the backscatter minimum, the term is better described as SOR (start of runoff = point of time, where water is starting to leave the snowpack and either penetrates into the ground or causes surface runoff underneath the snowpack), which is in line with Marin et al. 2020. |
| End of snowmelt (EOS) | End of snow cover (EOS) | 1. This term was not criticised by the reviewers and there is no interference with other publications, as this is the new variable identified by our study.
2. However, the term end of snowmelt is probably not optimal, because the time lapse images give only information about SC but not about melt.
3. "End of runoff" (EOR) is also not fitting, because no corresponding validation is available from the time lapse images.
4. What we actually try to detect is the "end of snow cover" and then visualize it in the snow cover depletion curves. Also only in that case validation with the time-lapse images makes sense.
5. EOS definitions: (SC fraction falls below 50 % for time lapse imagery; S-1 time series meets threshold/derivative condition). |

| | | |
|---|---|---|
| SOD (solely for time lapse imagery) | - | first observable decrease of SC fraction below 100 % in the time-lapse imagery for specific pixel. |
| EOD (solely for time lapse imagery) | - | point in time when SC fraction in the time-lapse imagery of a specific pixel reaches 0 %. |
| Perennial snow | End-of-season snow-covered | Reviewer 1 criticised the used terms, because they are not in line with the standard, as these areas might persist/be snow-free only for one single year. Hence, we renamed them to better describe the state actual being observed. |
| Permanently snow-free | Start-of-season snow-free | |

Thus, the following major changes have been made to the manuscript:

- A paragraph was added in section *2.1 Study area* to introduce the Kobbefjord Research Area (L. 112-126).
- A paragraph was added in section *3.1 Sentinel-1 snowmelt and snow cover products* to introduce the derivative approach (L. 195-208).
- The result section has been extended to include the results generated for the second site as well as for the second approach using the derivative (L. 319-326; 336-344; 349-356).
- The discussion has been extended to address new insights from the additional results as well as to address the important issues raised in your comments (L. 388-401; 423-432; 450-453; 464-473, 482-485).

Please note as well that we have changed the title of the manuscript, which we now think is more precise. Please note as well that an additional co-author was added. Kerstin Rasmussen from ASIAQ joined, as she has maintained the time-lapse cameras in Kobbefjord and as she is an expert for environmental setting in Kobbefjord. Additionally, we provide melt layers of the processed S-1 bursts as a supplement to show the entire output of the S-1 workflow and its capabilities beyond the smaller reference sites of the time-lapse imagery.

Yours sincerely and on behalf of all authors,

Sebastian Buchelt

**Point-to-point response:**

Dear reviewers,

The reviewer's comments, author's responses and the respective phrases changed in the new manuscript are highlighted in different colors for better separation as follows:

**Reviewer's comments: orange**
**Author's responses: green**
**Respective parts of the manuscript: blue**

In cases of generic comments or comments which address larger parts of the manuscript (e.g. adapted terminology; Kobbefjord site description ; derivative approach introduction ; Kobbefjord & derivative results; rearranged discussion), we apologize that we could not mention all changes here but we wanted this document to keep a reasonable amount of pages. All changes made can be found in the file highlighting the changes made since the submission of the preprint.
Thank you for your time and effort!

Yours sincerely and on behalf of all authors,

Sebastian Buchelt

**Review 1**

General comments

The paper studies a time series of Sentinel-1 images over Zackenberg valley, Greenland for snow melting and compares with high-res terrestrial optical data over an area. They use backscatter thresholds to identify start and end snowmelt and wet/dry/perennial snow status. They find that HV-pol data outperforms VV-data.

The paper uses S1 data at an interesting site and combine with high-resolution in situ data to validate their method development. This is valuable since S1 data is a very useful at high latitudes where optical data often fails due to cloud cover/darkness.

The authors suggest a method to retrieve snow cover (SC) at 20m spatial resolution. The method is based on finding the minimum backscatter for each pixel in the time series (SOS) where snow cover is expected to be 100% and subsequently the end of the melting season (EOS) when snow cover is 0% based on a fixed threshold 4dB above minimum. This method deviates significantly from the standard approach (Nagler&Rott,2000) relying on static reference images. A main obstacle is that the whole time-series need to be considered before SC-maps are made. Hence near-real time mapping is out of the question.

Yes, this is true. The approach is not suited for near-realtime mapping as the entire time series is needed for the estimation. Please note that the main objective of the study is the characterization of the spatiotemporal dynamics of snowmelt and related parameters.

It is somewhat unclear if the authors think that this approach is globally applicable, or only gives the best snow cover estimates for the current site. A discussion on the applicability of the method worldwide for various conditions (mountainous with variable local incidence angle, variable land types, forested areas etc.) would also be valuable. If the method has limited applicability outside Zackenberg, then why not only use the optical time series?

We agree with the reviewer that further assessment in different environments and locations is necessary. For this we have now included a second test site, Kobbefjord (low Arctic, West-Greenland). Further, we extended the discussion on the transferability, indicating that the method is developed based on the physical principle of SAR backscatter during snowmelt. Therefore it is independent from the test site as the physical principle is everywhere the same, e.g. the characteristic seasonal pattern has been observed in the Alps (Marin 2020 et al.) as well as for other areas around the globe (Lievens 2019 et al.). Parametrisation, however, must be adapted and adjusted to the local conditions. Further research is needed here, how the optimum parameters could be identified. Please note in this context that we also present an approach based on the derivatives of the time series. This approach favors transferability as it is self-adjusting and not linked to a fixed global (scene) threshold.

L. 466-471: "The derivative adapts well to different environmental settings (high & low arctic) and different seasonal conditions (high and low snow depths). Hence, the derivative approach could be applied to another site without requiring prior optimization of the threshold. Further, both approaches are based on the physical principle of SAR backscatter during snowmelt. Hence, they are expected to work well in other low vegetated areas as the characteristic seasonal pattern has been observed in the Alps (Marin et al., 2020) as well as around the globe (Lievens et al., 2019). Parametrisation, e.g. of the threshold or the melt season, however, must be adapted and adjusted to local conditions.

I dislike the fact that the authors only processed one satellite geometry. Unfortunately, there doesn't seem to be systematic acquisitions with S1 for more geometries (SciHub). This could have been used to confirm results and/or improve the temporal resolution for the SOS/EOS. Also, variability in incidence angles could shed some more insights. An alternative could have been to also look at EW mode data (HV) since Greenland is covered by numerous geometries, but at the cost of lower spatial resolution.

Yes, it is correct that only one satellite geometry provides data in IW mode for Zackenberg. We agree that, even though, LIA corrected $\gamma_0$ was used, an assessment of it could provide further detail on incidence angle dependence, however, we think this is right now out of the scope of the paper. Please consider that we now show results also for a second test site, having a different morphology. Further, an explicit goal of our investigation is to provide small-scale estimates of SC and related parameters and to study the temporal evolution of the SAR signal in relation to high-resolution in situ data, which provide snow cover fraction estimates. As such

we have not considered the use of EW data as these (i) will not allow to study the small-scale SC heterogeneity due to their coarse spatial resolution and (ii) are not suited for the rather small sizes of the test sites (42 and 7 km² covered by the in situ cameras). Nevertheless, we believe that EW data could be used with our approach for snowmelt detection and snow cover depletion mapping on larger scales with coarser resolution in future studies. We added a statement to address both issues in the discussion.

L.482-485: "If available, different S-1 orbits could be used to increase the temporal resolution of the product, but need to be analyzed separately due to differences in local incidence angle and acquisition times. Further, EW data could be used with our approach for snowmelt detection and snow cover depletion mapping on larger scales with coarser resolution."

Overall, I feel that the paper looks at a somewhat limited time series (one geometry, two years) with good results. Using the minimum backscatter per pixel/year is interesting, but I believe more work should be delivered to convince readers that this can be applicable to other sites/landscapes. The paper has limited value if it is only applicable to Zackenberg.

We agree that extending this case study to other sites is important. Therefore, we have included a second test site and now show results also for the Kobbefjord region (Western-Greenland close to Nuuk). The Kobbefjord research area is, like Zackenberg, part of the Greenland Ecosystem Monitoring programme. Therefore, it offers a similar setup and also time-lapse camera imagery of the valley is available. As indicated in the revised version, we have repeated the entire processing of the camera imagery and of the Sentinel-1 time series for the Kobbefjord test site and we present results of both regions. Note that the environmental setting in Kobbefjord (low Arctic) is different to the setting in Zackenberg (high Arctic), which is also evident when studying the SC and its temporal evolution. Even though, the presented methods (threshold- and derivative-based approaches) perform well for both sites and produce reliable estimates, which compare well with the in situ measurements. For sure this is not a proof for a truly "global applicability" (which is also outside the scope of the contribution), but results confirm that the general design of the approach is not over-fitted but transferable. Further, the approach is based on the physical principle of SAR backscatter during snowmelt, hence the method should be applicable elsewhere, as indicated above. Beside the proposed approach, we think that results gathered by the use of the high-quality in situ data on the snow cover fraction provide an interesting merit, as these provide insights on the temporal evolution of S-1 data for the SC and its depletion.

20 m pixels lead to significant speckle noise. Authors should evaluate whether slightly lower resolution (e.g. 50m, 100m) could lead to better performance in general.

The decrease in spatial resolution might in fact lead to better results, as a generalization will cause a better signal to noise ratio (reduction of variance) and a better radiometric stability (less speckle noise when increasing the number of looks). However, we wanted to make use of the high-resolution in situ camera imagery and the Sentinel-1 IW data. As such, we explore if it is possible to estimate SC and related parameters on comparable small-scale using S-1 time series and to infer effects related to the SC fraction cover during the melt, i.e. to observe/characterize this processes a high-spatial resolution is inherently required due to the patchiness of SC during the depletion.

Although terrain-corrected gamma is used, it could be interesting to look at local incidence angles. If the variability in incidence angles is large, there may be a variability in the contrast of gamma during wet/dry-conditions as noted by e.g. Nagler et al., 2018, which could lead to a more variable result with respect to whether VV or VH is the preferred polarization. High local incidence combined with wet snow could lead to signals close to NESZ. By eye measure from fig 2 I suspect that the incidence angle for the site is around 30 deg. This is close to the range where Nagler et al. (2018) also state that VH is superior to VV, and the results are hence supported. However, perhaps some of the poor classifications could be explained if higher local incidence angles are involved somewhere in the sloping terrain?

Thanks for pointing this out. While the analysis of the LIA is definitely of interest, we have decided not to address this issue in the revision: for Zackenberg, only one relative orbit is available (the Zackenberg Valles is located in the center of the second swath (IW2) and center incidence angle for IW2 is 38.7°), as such the possibilities to study LIA are very limited, also taking into account that our reference data is biased as the acquisition geometry of the camera prevents analysing all aspect and slope angles. For Kobbefjord more geometries (S-1) are available, however, here the same issues with respect to the camera position applies. Furthermore, snow distribution as well as vegetation and soil type depend on slope and aspect. Hence, identifying the individual contribution of the LIA would be difficult without considering these other effects in addition. A detailed assessment of all factors is, however, beyond the scope of our case study as well as is our study site too small to capture all potential effects. Nevertheless, we have included a statement that further research is required to assess the effect of LIA as well as snow properties, vegetation density and substrate on the threshold setting.

L.463-466: "Threshold setting must be assessed in more detail to confirm, whether a global threshold is applicable also in other sites and years, and to analyse which effects different snow properties, vegetation, substrate and local incidence angle might have on the seasonal backscatter behaviour of S-1 above snow, e.g. seasonal minimum & variation, as well as the resulting product accuracy."

Figure 5a: Colours/symbols used to separate between time laps based/in situ based data for 2017 and 2018 do not correspond with legend. Perhaps use different symbols for each year like fig 8?

Thank you for the suggestion. We adjusted that accordingly.

The term perennial snow is used throughout the paper. In my view this is snow patches that persists over several years, whereas the authors redefine it as snow that does not vanish over one summer. E.g. fig 9 shows significant amounts of perennial snow in 2018 but not in 2017. I think a better term should be found. E.g. Snow does not melt for the current year. The same could also be invented about permanently snow-free pixels. These, I suspect, are in general only snow free for the current season.

Thank you for the valuable suggestion regarding the terminology. We renamed the perennial snow class to end-of-season snow-covered and the permanently snow-free class to start-of-season snow-free to better describe the actual observation made. Please also see the general comment at the very beginning.

Technical matters

Line 374: temporal/spatial has been switched: should be … much higher temporal (1 to 10 days) and spatial (2.5m)… Perhaps also reconsider "much higher temporal" since S1 has 6 days temporal resolution? "Much" fits better on the spatial resolution.

Thank you for pointing that out. We adjusted it accordingly.

L.495: "much higher spatial (2.5 m) and higher temporal (1 to 10 days) resolution than the S-1 product"

**Point-to-point response:**

**Review 2**

**General comments**

This study has utilised 2 years of Sentinel-1 SAR data covering a small study area in northeast Greenland to develop a method for mapping snow cover based on the temporal the radar backscatter during snowmelt and high resolution snow cover fraction observations from time lapse imagery. Traditionally remote sensing of snow cover has relied on the use of optical sensors to detect snow but these methods are limited by cloud cover and during periods of low solar illumination which is a problem in Arctic areas where polar night is present for a part of the year. As such, an approach that can offer snow cover mapping capabilities under such conditions would be advantageous as well as a providing a complementary data source.

The approach claims to be able to map both dry and wet snow cover but based on the fact that the method relies on the relationship between SAR backscatter during the melt period (i.e. when snow cover is wet) and snow cover fraction information from high spatial resolution time-lapse imagery, I find it difficult to understand how the method can provide a solution for dry snow cover mapping. Detection of perennial snow and permanently snow free pixels suggests binary snow cover mapping, but this does not provide additional/improved information with respect to optical methods that can for example be used to derive snow cover fraction beyond the SOS/EOS period which is studied here.

Thanks for your comment on this. It is correct that the approach is focusing on the melt, as such we have highlighted this focus in the title (added "depletion"). Kindly also see our comments provided below and in the very beginning of the response.

Overall, the method seems very case study specific and it is unclear whether the same approach could be applied globally in other areas with seasonal snow cover. The time period of data acquisition (2 years) is also limited and the data presented suggests large variations in SAR backscatter can occur from year to year. However, I do believe the results are worth publishing but the content should be revised to reflect that the method has until now only been applied to a limited dataset. Moreover I don't think the method will be useful as a standalone method for snow cover mapping due to the limited part of the year on which the method is based (i.e. snowmelt), but can certainly complement existing methods.

We have addressed the issue of the transferability in two ways: First, we have now included an approach that uses the derivatives of the time series and, therefore, operates more adaptively. It is presented along with the threshold-based approach. Results point out that accuracies similar to the ones achieved from the threshold-based method can be realized. As now discussed, an approach using derivatives is believed to be less sensitive to signal-differences caused by different conditions of the snowpack or the land-cover. Therefore, the derivative-approach favors

transferability. Second, we have included a second test site and now show results also for the Kobbefjord region (Western-Greenland close to Nuuk). The Kobbefjord research area is, like Zackenberg, part of the Greenland Ecosystem Monitoring programme. Therefore, it offers a similar setup and also time-lapse camera imagery of the valley is available. As indicated in the revised version, we have repeated the entire processing of the camera imagery and of the Sentinel-1 time series for the Kobbefjord test site and we present results of both regions. Note that the environmental setting in Kobbefjord (low Arctic) is different to the setting in Zackenberg (high Arctic), which is also evident when studying the SC and its temporal evolution. Even though, the presented methods (threshold- and derivative-based approaches) perform well for both sites and produce reliable estimates, which compare well with the in situ measurements. For sure this is not a proof for a truly "global applicability" (which is also outside the scope of the contribution), but results confirm that the general design of the approach is not over-fitted but transferable.

Please note as well that the physical principle is the same, e.g. the characteristic seasonal pattern has been observed in the Alps (Marin 2020 et al.) as well as for other areas around the globe (Lievens et al.). Hence, they are expected to work well in other low vegetated areas. Parametrisation, however, must be adapted and adjusted to the local conditions. Further research is needed here, how the optimum parameters could be identified.

Beside the proposed approach, we think that results gathered by the use of the high-quality in situ data on the snow cover fraction provide an interesting merit, as these provide insights on the temporal evolution of S-1 data for the SC and its depletion.

**Specific comments**

Abstract l.12: "....enabling large-scale SC monitoring at high spatiotemporal resolution (20m, 6 days) with high accuracy" - this seems a somewhat bold claim given the small size of the case study area (45m2?) on which the method has been based.

Agree, we adjusted the statement accordingly. This is clearly a case study now comparing two test sites (low and high Arctic). Additionally, note that analysing the backscatter evolution over time does require high-quality reference data, which of course can only have a limited spatial extent.

L. 14-16: "Based on the physical principle of SAR backscatter during snowmelt, our approach is expected to work well in other low vegetated areas and, hence, could support large-scale SC monitoring at high spatiotemporal resolution (20 m, 6 days) with high accuracy."

Line 83: "...using adaptive thresholds" - the results would suggest that different thresholds have been tested but a fixed threshold of 4dB has actually been used to produce the snow cover maps using only the HV data. This statement needs revising.

Agree, we have removed the term adaptive from the statement. As indicated above we have also included a method based on the derivatives of the time series.

L. 85-86: "... using thresholds based on the seasonal minima of the SAR time series as well as the backscatter derivative for fast, simple, but effective SC mapping during snowmelt."

Figure 4 illustrates the specificity of the method. The workflow diagram is very involved considering the small size of dataset and case study area. Moreover it is difficult to follow. A new method for estimating snow cover that uses SAR ought to be more generic to be of use elsewhere. The authors do not specify whether this was the objective of the work, or if the goal was simply to develop an approach which could be used solely for the area of interest.

As recommended by the first reviewer, we included a part in the discussion to explain why we think this method is applicable also elsewhere, but that parameterization might be site specific. Please also note our statement on major issues in the beginning. Besides, we splitted Figure 4 for better readability into two parts as recommended by Reviewer 4: the first part shows the Sentinel-1 workflow and the second shows the workflow for the reference data acquired from the time-lapse imagery.

Lines 280-281: "... the t values for the best results dependent on the used polarization and the observed year" - suggests the need for an adaptive threshold to deal with the seasonal variations in backscatter due to for example, snow depth as alluded to in the discussion.

We agree, an adaptive threshold could produce better results and should be studied in further research. We believe that especially in landcover types with strong SAR backscatter events driven by SC and low effects by other land surface processes, this could have great potential. As indicated above, and as also suggested by other reviewers, we have addressed this issue by introducing a method based on the derivatives of the time series.

Respective paragraph in the discussions addressing the last two comments:

L. 463-471: "Threshold setting must be assessed in more detail to confirm, whether a global threshold is applicable also in other sites and years, and to analyse which effects different snow properties, vegetation, substrate and local incidence angle might have on the seasonal backscatter behaviour of S-1 above snow, e.g. seasonal minimum & variation, as well as the resulting product accuracy. The derivative adapts well to different environmental settings (high & low arctic) and different seasonal conditions (high and low snow depths). Hence, the derivative approach could be applied to another site without requiring prior optimization of the threshold. Further, both approaches are based on the physical principle of SAR backscatter during snowmelt. Hence, they are expected to work well in other low vegetated areas as the characteristic seasonal pattern has been observed in the Alps (Marin et al., 2020) as well as around the globe (Lievens et al., 2019). Parametrisation, e.g. of the threshold or the melt season, however, must be adapted and adjusted to local conditions."

Lines 366-367: "...could be used for a holistic hydrological monitoring of SC from the scale of a single catchment up to pan-Arctic observations" - as highlighted earlier, I think this kind of statement/claim is somewhat bold, given that the method has been developed using such a small area of study and only two years of data. Furthermore the method has not been demonstrated on areas elsewhere. This claim should be revised to something more realistic and which reflects the results of a dataset which is limited in both spatial and temporal dimensions.

Thanks for pointing this out. We have revised the statement. In addition please note that a second test site was added and an approach using the derivatives of the time series was introduced.

L. 486-487: "...could support in combination with other methods and datasets a holistic hydrological monitoring of SC from the scale of a single catchment up to pan-Arctic observations.

Lines 386-387: "The optimal seasonal threshold value increases in accordance with snow depth" - I do not recall any snow depth data having been presented in the study, so this statement seems rather speculative until supported by data analysis.

The only parameter that allows for a spatial assessment is the SC fraction cover data that we computed using the camera imagery. Higher SD is an observation made by another source, which we mention in the discussion. But indeed, no pixel based SD vs. threshold comparison could be made. Hence, it might be a possible explanation for the increase in the threshold, but is not assured. Therefore, we have removed the statement.

**Point-to-point response:**

**Review 3**

The paper presents interesting findings on the behavior of C-band SAR backscatter during snowmelt over a region in northeast Greenland, introduces a novel algorithm to map snow cover and validates the results with snow cover derived from time-lapse images. In my opinion, this work is very relevant, timely and novel. In particular, a novelty is the focus on the mapping of the (wet) snow-covered area/fraction after the backscatter reaches its seasonal minimum until snow-free conditions. Most previous approaches have focused on detecting the (more early) onset of liquid water present in the snowpack (by thresholding the decrease in backscatter relative to a dry snow reference). There are however a number of aspects that could still be further improved.

Major/general comments:

One general concern is that the authors refer to the detected snow before the minimum in backscatter is reached as dry snow. Similarly, the minimum backscatter is defined as the start of snowmelt (SOS). However, before the minimum backscatter is reached, there is likely already a substantial amount of liquid water present in the snowpack (which causes the low backscatter), meaning that the snow can arguably not be defined as dry. Note that most other wet snow detection algorithms classify wet snow before the minimum is reached (as soon as backscatter becomes lower than a certain threshold below a dry snow reference). I'd recommend that early in the paper, the authors clearly specify what is meant here with dry/wet snow and SOS, and frame this in accordance with the literature. I'd furthermore recommend not to focus at all on the transition from dry to wet snow (also because this is not, and cannot be, validated with the available reference data as appropriately identified by the authors), and thus to limit the methods, analysis and discussion to the detection of wet snow-cover fraction (which I also believe is the main novelty of the paper).

Thank you for raising these important issues. We agree that the terminology should be revised and did so. Please note our statement on this in the very beginning of the response where we outline the new terminology and that we have changed the title of the manuscript highlighting that the focus is on the snow cover depletion. Further, no differentiation between wet and dry snow is made anymore, as this is not the focus of our case study.

L73: "the snowpack is nearly transparent to C-Band SAR": Please modify this statement. There is considerable evidence that S-1 C-band observations, especially in cross-polarization, show an increase in scattering from dry snow accumulation by several dB. Accordingly, Fig. 1 should be modified to illustrate this increase in backscatter with dry snow accumulation. Also, the snow-free ground backscatter in cross-pol can sometimes (maybe not in this study region) be lower than the dry snow-covered backscatter in winter (depending on soil and vegetation conditions). This could also be mentioned.

Thanks for your statement on this. You are correct, we have adjusted Figure 1 accordingly (increase during accumulation; potential lower snow-free maxima). The increase in cross-polarization is mentioned now, even though hardly observable in our study site (1-2 dB maximum and barely above the observed variability of the signal at the spatial resolution we use (20m).

L.74-76: "During winter, the liquid water content is close to 0 % and the snowpack is nearly transparent to C-Band SAR, unless for larger snowpacks (>1 m), where a sensitivity of C-band cross-polarization is indicated by Lievens et al. (2019, 2021).

Some of the main problems of the study are the over-detection of perennial snow-free (because the threshold is too high for the more limited backscatter increase), and at the same time an under-detection of snow cover in areas where the threshold is too low. I'd strongly recommend to test using the derivative of backscatter over time to identify the EOS, rather than using a fixed 4 dB threshold. The increase in backscatter towards EOS may depend on the amount of liquid water in the snow, and the substrate and vegetation conditions, among other aspects. Thresholding the low derivative (low change in backscatter over time, following a strong increase) could potentially solve these issues, and may allow to find a more accurate EOS both in cases for which now smaller and larger thresholds (than 4dB) would ideally be needed.

Thank you for your suggestions on this. We followed your advice and we have now included an approach that uses the derivatives of the time series and, therefore, operates more adaptively. It is presented along with the threshold-based approach. Results point out that accuracies similar to the ones achieved from the threshold-based method can be realized. As now discussed, an approach using derivatives is believed to be less sensitive to signal-differences caused by different conditions of the snowpack or the land-cover. Therefore, the derivative-approach favors transferability.

Furthermore, it would be interesting to further investigate the regional differences in backscatter increase from the minimum towards the EOS, to determine its drivers. Is it mainly linked to snow depth, to the substrate type, topography, or vegetation?

While this is definitely an interesting point, we have decided not to address this issue in the revision: for Zackenberg, only one relative orbit is available (the Zackenberg Valles is located in the center of the second swath (IW2) and center incidence angle for IW2 is 38.7°), as such the possibilities to study LIA are very limited, also taking into account that our reference data is biased as the acquisition geometry of the camera prevents analysing all aspects and slopes. For Kobbefjord more geometries are available, however, here the same issues with respect to the camera position applies. Furthermore, snow distribution as well as vegetation and substrate type are also dependent on slope and aspect. Hence the contribution of each individual factor would be difficult to distinguish without considering the other effects. A detailed assessment of all factors is beyond the scope of our case study as well as is our study site too small to capture all potential effects. However, we mentioned the required need for further assessment in the discussion.

L.463-466: "Threshold setting must be assessed in more detail to confirm, whether a global threshold is applicable also in other sites and years, and to analyse which effects different snow properties, vegetation, substrate and local incidence angle might have on the seasonal backscatter behaviour of S-1 above snow, e.g. seasonal minimum & variation, as well as the resulting product accuracy."

Specific comments:

Please mention upfront in the paper that the algorithm proposed here is empirical.

Thank you for this suggestion. We have added this information.

L.9: "We here present a novel empirical approach based on the temporal evolution of the SAR signal …"

L110: The use of S-1 observations in extended wide swath mode would have resulted in much denser time series (which in the discussion you state as a potential pathway for improving the results). Is there a specific reason why the IW mode was used? You could consider repeating the analysis with EW data (and simultaneously also investigate the combination of different orbits).

An explicit goal of our investigation is to provide small-scale estimates of SC and related parameters and to study the temporal evolution of the SAR signal in relation to high-resolution in situ data, which provide snow cover fraction estimates. As such we have not considered the use of EW data as these (i) will not allow to study the small-scale SC heterogeneity due to their coarse spatial resolution and (ii) are not suited for the rather small sizes of the test sites (42 and 7 km² covered by the in situ cameras). Nevertheless, we believe that EW data could be used with our approach for snowmelt detection and snow cover depletion mapping on larger scales with coarser resolution in future studies.

We have further just used one relative orbit (IW), as acquisition geometry needs to be constant throughout the time series to ensure a comparability of the measurements. Nevertheless, we point out in the discussion that additional orbits (IW), if available, might be used to densen the time series; however, different orbits need to be analyzed separately, due to differences in local incidence angle and, more importantly, different acquisition times, which need to be considered. It might then be best to analyse each geometries time series individually instead of merging all data in one time series.

We added respective comments to the discussion:

L.482-485: "If available, different S-1 orbits could be used to increase the temporal resolution of the product, but need to be analyzed separately due to differences in local incidence angle and acquisition times. Further, EW data could be used with our approach for snowmelt detection and snow cover depletion mapping on larger scales with coarser resolution."

L110: Is there a reason why SLC S-1 images were selected rather than GRD images (which would have simplified the processing)?

This is just due to practical reasons, we first intended to add polarimetric features as well as coherence into the analysis, but then decided to stick with this simple and fast thresholding approach. Further, we have processed the data to gamma nought intensity using the terrain flattening approach implemented in SNAP and using the high-resolution Arctic DEM. As such we expect to have better radiometry and geolocation compared to the GRD product.

L176: "The threshold of 9 dB is set in accordance with our observations that HV snow-free summer intensity does not exceed the seasonal minimum by more than this value": this may well be the case in your study area, but in other regions (for instance prone to significant snow accumulation and melting, and/or followed by vegetation growth), I can imagine the increase can easily be larger than 9 dB. Please clearly state that this criterium may potentially not apply to different regions.

Thanks for pointing this out. We have modified the statement accordingly.

L.219-220: "The threshold of 9 dB is set in accordance with our observations that HV snow-free summer intensity does not exceed the seasonal minimum by more than this value, but might not apply for other sites."

Figure 4 indicates that backscatter data are analyzed from March to August, whereas Figure 3 indicates May to October?

Seasonal minimum is searched within this period (March-August). For Zackenberg, In 2017 the search is limited only starting from May due to the limited dual-pol timeseries. As no camera images are available (Zackenberg) or used due to mountain shadow (Kobbefjord) before May and after October / September for comparison, the range is not shown even though Sentinel-1 images of the entire year are used.

L227: "areas with SOD before 1 June are excluded": how many areas are excluded by this criterium?

Thank you for the specific question regarding the exclusion. After cross-checking our analysis, we realized that this exclusion was unnecessary. Solely start-of-season snow-free areas are now excluded. However, the results and interpretation have not changed, as the wrong exclusion due to our mistake had no significant influence on them. We have adjusted the statement accordingly.

L.271-272: "Thus, areas with EOD after 15 August and *start-of-season snow-free* areas are excluded from analysis."

L230: "Areas with 100 % and 0 % SC fraction were further segmented according to the temporal distance to SOD and EOD, respectively": this statement is not clear to me.

Thank you for pointing that out. We rephrased the mentioned sentence to clarify this step in our analysis.

L.274-277: "Furthermore, we identified the temporal distance to SOD and EOD for areas with 100 % and 0 % SC fraction, respectively, in order to capture the development of backscatter intensities before SOD and after EOD. Negative days indicate the number of days before the first observable decrease in SC fraction (SOD). Positive days indicate the number of days after the observed snow cover fraction has reached 0 % (EOD)."

L234: I'm not sure how useful the comparison of your threshold with that of Nagler's approach is. The latter refers to the decrease in backscatter relative to a dry snow reference, which is taken somewhere in summer to early winter (and it is not very clear when to best take the reference and what the impact of that timing is). Your threshold is used for the increase in backscatter after reaching the backscatter minimum. Furthermore, I don't think Nagler used gamma nought, and also focused on a different region (with potentially contrasting conditions in substrate, vegetation and snow properties). This should at least be mentioned if the authors still wish to include this comparison.

Thank you for this valuable comment. We mentioned the systematic differences between the two approaches now. Due to these differences, a detailed assessment of the thresholds is, in our opinion, even more important.

L. 279-283: "According to our observations, the selection of polarization and threshold t (Eq. (2)) is crucial for the accuracy of the S-1 snow products of the threshold-based approach. Using the standard threshold (2 to 3 dB) in Nagler's method (Nagler and Rott, 2000; Nagler et al., 2016; Snapir et al., 2019) might not be suited due to the different threshold basis (snow-free/dry snow backscatter vs. seasonal minimum) and the use of different levels of preprocessed SAR data ($\sigma^0$ vs. terrrain-corrected $\gamma^0$). Therefore, we investigated a threshold range from 2 to 8 dB for HV and from 2 to 10 dB for HH polarization."

Is there a difference between EOS and EOD, and SOS and SOD? Please clarify.

As noted above, we addressed this issue and generally revised the terminology.

SOS: renamed as start of runoff (SOR), only detectable by S-1 time series.

EOS: end of snow cover (SC fraction falls below 50 %; S-1 time series meets threshold/derivative condition).

SOD: solely derived from the time lapse imagery, defined as: first observable decrease of SC fraction below 100% in the time-lapse imagery for specific pixel.

EOD: solely derived from the time lapse imagery, defined as: point in time when SC fraction in the time-lapse imagery of a specific pixel reaches 0 %.

Figure 6: Is this figure derived from averaging backscatter in space or just for a single location (I may have overlooked that in the description). In case of the former, how many pixels were included, and how does the averaging impact the trends. Also, was averaging performed in linear or dB scale?

Thank you for the comment. We collected all SC fraction values and backscatter intensity values of same-day acquisitions and rearranged them according to their respective SC fraction value. Furthermore, we identified the temporal distance to SOD and EOD for areas with 100 % and 0 % SC fraction, respectively, in order to capture the development of backscatter intensities before SOD and after EOD. Negative days indicate the number of days before the first observable decrease in SC fraction (SOD). Positive days indicate the number of days after the observed snow cover fraction has reached 0 % (EOD). Then the collected backscatter intensities were grouped by their respective SC fraction or temporal distance to SOD or EOD and averaged in dB scale.

Figure 6: It would have been interesting to see the full yearly S-1 backscatter timeseries (instead of only the period from Spring/May onwards). Perhaps the full time series could also give insight on the snow depth, which could be helpful for defining the thresholds in the snow-cover detection. How deep is the snowpack typically in that area? Further, the full timeseries would reveal if for instance the autumn/winter backscatter is also relatively lower or higher compared to the end of spring and summer backscatter. Now, it seems the maximum backscatter is always obtained in summer. Would that be primarily caused by the vegetation, or is the snowpack already wet (i.e., liquid water present in the snowpack) at the start of your analysis? I'd expect another minimum backscatter in autumn and an increase during winter (depending on the snow depth). Is this behavior observed in your study region?

Thank you for raising this interesting point. First, snow-free summer backscatter intensity is (in our study sites) generally above the winter backscatter values (co-pol and cross-pol) probably due to the effects of vegetation growth. We do see a decrease and a minimum of backscatter values in autumn, but usually they do not reach values as low as during spring, except for a few spots in single acquisitions. We assume that in such cases we have events of wet snowfall or snow on unfrozen ground which cause these low backscatter values similar to wet snow.

According to our observations, backscatter during winter shows hardly any changes. There might be an increase in cross-pol backscatter of 1-2 dB maximum, but it is really low and does barely exceed the observed variability of the signal at the spatial resolution we use (20m). Only for end-of-season snow-covered areas as well as areas covered by glaciers (outside the camera field of view) we observe a strong increase with refreeze. We conclude that there might be a chance that a backscatter increase along with increasing snow depth is observable, but is very limited due to lower snow depths (1.4m max) compared to observations made by Lievens et al. 2019, 2021). Further, also other effects not related to snow accumulation such as episodic snowmelt could strongly affect the backscatter (i.e. an increase in number and size of grains and ice lenses) which need to be considered.

The Figures below show the backscatter time series for 2000 randomly selected pixels:

[Figure]

L273-275: As mentioned above, classification by thresholding the change in backscatter over time (derivative) instead of the absolute intensity could potentially help with this.

Thanks for your suggestion on this. Please see our answer to your previous remark on the derivatives.

L285: Would the authors also have a physically-based hypothesis on why the use of HV may be better than HH?

That performance is potentially linked to the local incidence angle. As noted by Nagler et al., 2018 contrast between wet and dry conditions in HH and HV is influenced by the local incidence angle. Results here support Nagler et al., 2018, which have found VH superior to VV.

L287: The overall accuracy can be strongly impacted by the number of 0's and 1's. Have you tested for instance the Cohen's Kappa metric (also based on the classification matrix) which is less impacted by the numbers of positives and negatives?

Thank you for the comment. We agree that Cohen's Kappa metric is better suited to measure performance of imbalanced datasets. In our case however, the Kappa Metric would generate low values for the periods, when the entire area is either fully snow covered or fully snow-free, as low sensitivity and specificity for the minority classes occur. In contrast, the most relevant time period to detect SC correctly is during high rates of snowmelt. As such, we decided to exclude Kappa and displayed the main results with a focus on the FP and TP rate, the ROC respectively.

L290-291: The authors mention that a high FP is due to the undetected snowfall on DOY177, but wouldn't this classify as a false negative (i.e. no snow detected by S-1, but snow on ground in reality)? Further, have you looked at how this late snowfall is impacting the backscatter? I'd expect it could increase the scattering if snowfall is dry, or decrease scattering if snowfall is wet?

In this case we had a wet snow fall event decreasing the backscatter. Our approach detects only seasonal decrease and misclassified the area as snow covered before the event, which was not the case in the previous acquisition. Hence in this previous acquisition the area is not covered by snow, but classified as such, which results in an increased FP.

L305: The paper by Lievens et al (2019) mostly focuses on the increase in backscatter during dry snow accumulation (during autumn to early winter). This is a different time period, which also shows generally an increase in snow scattering, but likely due to different phenomena (for instance, there the increase is likely not caused by an increase in snow surface roughness). The reference to Marin is more appropriate here.

Yes, true. We adapted this sentence accordingly. Nevertheless, when looking into the timeseries plots we can observe the same specific pattern in their observations, even though they took it not in use.

L.384-386: "We observe a distinct seasonal behaviour in S-1 C-band backscatter with a clear decrease during early melt, reaching a minimum just prior to the onset of melting, followed by a constant linear increase towards the EOS. This is consistent with findings in Marin et al. (2020)."

L309: I'm not sure how much volume scattering there would still be. At this stage, the snowpack is very wet, and penetration (necessary for volume scattering) should be rather limited. I'd therefore expect that the increase in HV is caused by surface scattering with increased depolarization by the rougher snow surface.

We thank the reviewer for this very helpful explanation. We adjusted our statement accordingly.

L. 393-394: "The increase before SOD in HH is probably caused by the higher surface roughness, while increased depolarization at this rougher snow surface increases the backscatter intensity in HV."

L316: I think it might also be worth stating that your approach focuses on a different phase of the wet snow season than Nagler's approach (which is more focused on the initial detection of liquid water in the snowpack by the decrease in backscatter). So both approaches could be complementary.

Thank you for highlighting the potential complementary use. We added this in the discussion.

L. 477-479: "Potentially, the snow phase detection algorithm by Marin et al. (2020) could be incorporated to further separate melt phases in more detail and Nagler's method (Nagler and Rott, 2000) could be used to identify the start of wet-snow phase."

L334: The fact that no independent validation is performed should definitely be mentioned earlier in the paper and be identified as a shortcoming of the analysis.

We agree with you that this should be mentioned earlier and have addressed that. Also note that the approach using the derivatives (which was included in the revision, see above) is less susceptible to this issue.

L. 294-298: "Further analysis is carried out for the derivative approach as well as for the global polarization-threshold configuration with the best ROC (highest TP rate + lowest FP rate). The threshold-based product assessment is considered weaker as it conducted with the same reference used for identifying this configuration whereas no a priori knowledge is required for the derivative-based products."

L338: I'm not sure if different orbits (with different incidence angles and different timings when combining ascending and descending tracks) is likely to increase the performance. The impact of the incidence angle should be further investigated (or at least recommended for future research). Denser time series would be available when switching to the EW mode over Greenland.

Thank you for raising that point. The use of different orbits would be beneficial to densen the time series; however, different orbits would need to be analyzed separately, due to differences in local incidence angle and acquisition times. We will point that out more precisely in the discussion. Besides, we definitely recommend to investigate the influence of local incidence angle as well as snow properties, vegetation and substrate on the proposed threshold method for further research, as this would give further insights into the contribution of each parameter on the S-1 signal; however, it is out of the scope of our case study. Regarding the use of EW data, due to our goal to investigate small-scale SC and related parameters and to study the temporal evolution of the SAR signal in relation to high-resolution in situ data, we have not considered the use of EW data as these (i) will not allow to study the small-scale SC heterogeneity due to their coarse spatial resolution and (ii) are not suited for the rather small sizes of the test sites (42 and 7 km² covered by the in situ cameras). Nevertheless, we believe that EW data could be used with our approach for snowmelt detection and snow cover depletion mapping on larger scales with coarser resolution in future studies.

L.463-466: "Threshold setting must be assessed in more detail to confirm, whether a global threshold is applicable also in other sites and years, and to analyse which effects different snow properties, vegetation, substrate and local incidence angle might have on the seasonal backscatter behaviour of S-1 above snow, e.g. seasonal minimum & variation, as well as the resulting product accuracy."

L.482-485: "If available, different S-1 orbits could be used to increase the temporal resolution of the product, but need to be analyzed separately due to differences in local incidence angle and acquisition times. Further, EW data could be used with our approach for snowmelt detection and snow cover depletion mapping on larger scales with coarser resolution."

L345: Please mention that your method focuses on the transition of wet snow to snow-free, whereas the other referred approaches mostly identify the transition from dry to wet snow.

Thank you for raising this point. We have adapted the title of the manuscript accordingly and added "depletion" and also express more precisely that our approach detects SC (wet and dry, no separation made) solely during the melt period.

L358: Again, I find the wording of SOS ambiguous. There will already be much liquid water within the snowpack by the time the backscatter reaches its minimum value. Thus, one could argue that snowmelt starts much earlier in reality.

Thank you for highlighting this issue. Please note our statement on this in the very beginning of the response where we outline the new terminology. Start of snowmelt (SOS) is now renamed as start of runoff (SOR), which is in line with Marin 2020 et al.

L361: I'm not sure you can say that dry versus wet snow is detected by the algorithm (only wet).

Thank you for pointing that out. We adapted our terminology and do not separate between wet and dry snow cover anymore.

L. 474-476: "With this new approach, many relevant parameters for SC monitoring are detected at a weekly basis by the here proposed approach: State and extent of SC during melt, end-of-season SC and start-of-season snow-free areas. Further, important hydrological measures like start of runoff (SOR) and end of snow cover (EOS) are derived …"

L364: The literature on SWE reconstruction could be improved (e.g. mention the work of Margulis, Bair, etc.)

Thank you for this suggestion. We incorporated more literature here.

L. 479-481: "Provided at a spatial resolution of 20 m, hydrological models could further use this information to derive additional parameters like snow water equivalent (based on reconstruction approaches presented e.g. by Molotch and Margulis (2008); Kerr et al. (2013); Bair et al. (2016); Rittger et al. (2016)) …"

**Point-to-point response:**

**Review 4**

This study focus on two years of Sentinel-1 data covering a small part of the Zackenberg valley in northeast Greenland to develop an algorithm for mapping snow evolution during the melting season. This time series is compared with snow cover fraction observations from time lapse imagery. The physical background on which the proposed approach is based is already described in the literature and this work can be view as an interesting extension and validation of the findings by Marin et at. 2020. However there are still some aspects that should be further improved/clarified for being of interest to the scientific community.

General comments/concerns:

The title and the definitions are misleading w.r.t. the content of the paper (at least to me in the present form). In this context, the gamma nought time series can be exploited to find a time series of maps indicating snow status related to the loss of snow mass i.e., depletion curve. This is a important variable that can be extracted only with SAR information (differently from the snow cover depletion curve). However the definitions should be better described for avoiding confusion (making use also of fig. 1).

Thank you for this very useful comment. We agree that the terminology should be revised and also made use of Figure 1 to better introduce and describe the used terms. Please note our statement on this in the very beginning of the response where we outline the new terminology. As well also note that we have changed the title of the manuscript highlighting that the focus is on the depletion.

The use of only one track is limiting the understanding of the operational applicability of the proposed algorithm. Reconstructing the depletion curve with a sub-weekly sampling could be relevant in different contexts. This should be better analyzed and discussed.

Please see our comment in the very beginning of the response. We have just used one relative orbit (IW), as acquisition geometry needs to be constant throughout the time series to ensure a comparability of the measurements. Nevertheless, we point out in the discussion that additional orbits (IW), if available, might be used to densen the time series as indicated by you; however, different orbits need to be analyzed separately, due to differences in local incidence angle.

L.482-485: "If available, different S-1 orbits could be used to increase the temporal resolution of the product, but need to be analyzed separately due to differences in local incidence angle and acquisition times. Further, EW data could be used with our approach for snowmelt detection and snow cover depletion mapping on larger scales with coarser resolution."

The thresholds adopted in the paper are derived from the dataset from which the reference information is available. The two considered years, which show different    characteristics, already show a relative large variance in the results. Another independent test site(s) is

necessary to fully understand the scope of applicability of the proposed algorithm. The use of fixed thresholds, even interesting to demonstrate the method, limit the generalization of the algorithm especially when more advanced methods are available.

Thanks for commenting on this. We have addressed this issue twofold: First, we have now included an approach that uses the derivatives of the time series and, therefore, operates more adaptively. It is presented along with the threshold-based approach. Results point out that accuracies similar to the ones achieved from the threshold-based method can be realized. As now discussed, an approach using derivatives is believed to be less sensitive to signal-differences caused by different conditions of the snowpack or the land-cover. Therefore, the derivative-approach favors transferability. Second, we have included a second test site and now show results also for the Kobbefjord region (Western-Greenland close to Nuuk). The Kobbefjord research area is, like Zackenberg, part of the Greenland Ecosystem Monitoring programme. Therefore, it offers a similar setup and also time-lapse camera imagery of the valley is available. As indicated in the revised version, we have repeated the entire processing of the camera imagery and of the Sentinel-1 time series for the Kobbefjord test site and we present results of both regions. Note that the environmental setting in Kobbefjord (low Arctic) is different to the setting in Zackenberg (high Arctic), which is also evident when studying the SC and its temporal evolution. Even though, the presented methods (threshold- and derivative-based approaches) perform well for both sites and produce reliable estimates, which compare well with the in situ measurements. For sure this is not a proof for a truly "global applicability" (which is also outside the scope of the contribution), but results confirm that the general design of the approach is not over-fitted but transferable. Please note as well that the physical principle is the same, hence the method should be applicable elsewhere, as indicated above. Beside the proposed approach, we think that results gathered by the use of the high-quality in situ data on the snow cover fraction provide an interesting merit, as these provide insights on the temporal evolution of S-1 data for the SC and its depletion.

Specific comments:

Fig.1: A slight increase of LWC can produce a high decrease in the backscattering. So it's unreal that in the moistening phase, cycles of increase and decrease LWC are not influencing the recorded backscattering. Moreover, the decrease in SCF does not correspond to the runoff onset especially for deep snowpack. Interestingly this curve was introduced for high alpine snowpacks. Do you have SWE measurements showing that the runoff onset is correctly identified by your time series? This would be an interesting extension of the paper by Marin et. al.

Thank you for raising these important and interesting points. Figure 1 was adapted accordingly to your suggestion.

We agree that SWE measurements as a tool to assess the accuracy and meaning of SOR would be of great interest. However, our interest was to detect SC extent and its depletion as well as to assess the interaction between backscatter and small-scale SC fraction. So we decided to exclude SWE data. Further research could use available data from permanently

installed Snow Pack Analyzers, best case for more years to confirm the correct detection of SOR observations.

Fig. 4 is rather difficult to be read. I suggest to divide it for section 3.1 and 3.2.

Thank you for this good suggestion. We adapted it accordingly and split Figure 4 in two parts.

The works of Lievens et al. on snow depth retrieval, even if showing the characteristic melting curve, state an increase in the backscattering due to snow accumulation that seems not to be noticed in your case given the comment of line 73. It would be interesting to know if you find the same behaviour in your experimental analysis and in case provide a comment on this aspect.

Another reviewer criticised that such an increase is not mentioned here. We included it now. According to our observations, backscatter during winter shows hardly any changes. There might be an increase in cross-pol backscatter of 1-2 dB maximum, but it is really low and does barely exceed the observed variability of the signal at the spatial resolution we use (20m). Only for end-of-season snow-covered areas as well as areas covered by glaciers (outside the camera field of view) we observe a strong increase with refreeze. We conclude that there might be a chance that a backscatter increase along with increasing snow depth is observable, but is very limited due to lower snow depths (1.4m max) compared to observations made by Lievens et al. 2019, 2021). Further, also other effects not related to snow accumulation such as episodic snowmelt could strongly affect the backscatter (i.e. an increase in number and size of grains and ice lenses) which need to be considered.

We decided to not provide a comment on this due to the fact that our case study is focussing on a different time period of the year (melt season and not winter) and due to the limited number of winter observations (only one entire winter season (2017/18) available for each site due to the setup of our case study).

Line 140: Small et al. in their last paper (https://ieeexplore.ieee.org/document/9352976) comment that the implementation of their terrain flattening in SNAP is not correct. This should be better commented in the paper.

Thank you for raising this issue. We were not aware of this and we mention this now in the manuscript.

L. 168-169: "... and calibration to backscatter coefficient $\gamma^0$ using the terrain flattening approach similar to Small (2011) (Small et al., 2021) with the ArcticDEM (Porter et al., 2018) was applied.

Line 305-310: do you have any measurements showing the increase in the superficial roughness?

We agree with you that this would be very helpful to understand the underlying causes for the backscatter increase during runoff but, unfortunately, we do not have such data available for either of the two sites.